# T follicular helper 17 (Tfh17) cells are superior for immunological memory maintenance

Xin Gao[1,2], Kaiming Luo[2], Diya Wang[3], Yunbo Wei[4], Yin Yao[5], Jun Deng[2], Yang Yang[6], Qunxiong Zeng[2,7], Xiaoru Dong[3], Le Xiong[7], Dongcheng Gong[2], Lin Lin[8], Kai Pohl[1], Shaoling Liu[9], Yu Liu[9], Lu Liu[10], Thi HO Nguyen[11], Lilith F Allen[11], Katherine Kedzierska[11], Yanliang Jin[9], Mei-Rong Du[10], Wanping Chen[3], Liangjing Lu[7], Nan Shen[2,7], Zheng Liu[5], Ian A Cockburn[1]*, Wenjing Luo[3]*, Di Yu[1,6,12]*

[1]Immunology and Infectious Disease Division, John Curtin School of Medical Research, The Australian National University, Canberra, Australia; [2]China-Australia Centre for Personalised Immunology, Renji Hospital, School of Medicine, Shanghai Jiao Tong University, Shanghai, China; [3]Department of Occupational and Environmental Health and the Ministry of Education Key Lab of Hazard Assessment and Control in Special Operational Environment, School of Public Health, Fourth Military Medical University, Xi'an, China; [4]Laboratory of Immunology for Environment and Health, Shandong Analysis and Test Center, Qilu University of Technology, Shandong Academy of Sciences, Jinan, China; [5]Department of Otolaryngology-Head and Neck Surgery, Tongji Hospital, Tongji Medical College, Huazhong University of Science and Technology, Wuhan, China; [6]Frazer Institute, Faculty of Medicine, University of Queensland, Brisbane, Australia; [7]Department of Rheumatology, Shanghai Institute of Rheumatology, Renji Hospital, School of Medicine, Shanghai Jiao Tong University, Shanghai, China; [8]Department of Laboratory Medicine, Ruijin Hospital, School of Medicine, Shanghai Jiao Tong University, Shanghai, China; [9]Shanghai Children's Medical Centre, Shanghai Jiao Tong University, Shanghai, China; [10]Obstetrics and Gynecology Hospital of Fudan University (Shanghai Red House Obstetrics and Gynecology Hospital), Shanghai, China; [11]Department of Microbiology and Immunology, Peter Doherty Institute for Infection and Immunity, University of Melbourne, Melbourne, Australia; [12]Ian Frazer Centre for Children's Immunotherapy Research, Children's Health Research Centre, Faculty of Medicine, University of Queensland, Brisbane, Australia

*For correspondence:
ian.cockburn@anu.edu.au (IAC);
luowenj@fmmu.edu.cn (WL);
di.yu@uq.edu.au (DY)

**Competing interest:** The authors declare that no competing interests exist.

**Abstract** A defining feature of successful vaccination is the ability to induce long-lived antigen-specific memory cells. T follicular helper (Tfh) cells specialize in providing help to B cells in mounting protective humoral immunity in infection and after vaccination. Memory Tfh cells that retain the CXCR5 expression can confer protection through enhancing humoral response upon antigen re-exposure but how they are maintained is poorly understood. CXCR5+ memory Tfh cells in human blood are divided into Tfh1, Tfh2, and Tfh17 cells by the expression of chemokine receptors CXCR3 and CCR6 associated with Th1 and Th17, respectively. Here, we developed a new method to induce Tfh1, Tfh2, and Tfh17-like (iTfh1, iTfh2, and iTfh17) mouse cells in vitro. Although all three iTfh subsets efficiently support antibody responses in recipient mice with immediate immunization, iTfh17 cells are superior to iTfh1 and iTfh2 cells in supporting antibody response to a later

immunization after extended resting in vivo to mimic memory maintenance. Notably, the counterpart human Tfh17 cells are selectively enriched in CCR7$^+$ central memory Tfh cells with survival and proliferative advantages. Furthermore, the analysis of multiple human cohorts that received different vaccines for HBV, influenza virus, tetanus toxin or measles revealed that vaccine-specific Tfh17 cells outcompete Tfh1 or Tfh2 cells for the persistence in memory phase. Therefore, the complementary mouse and human results showing the advantage of Tfh17 cells in maintenance and memory function supports the notion that Tfh17-induced immunization might be preferable in vaccine development to confer long-term protection.

## Editor's evaluation

In regard to subpopulations of memory Tfh cells, the authors nicely showed the significance of Tfh17 cells. Although some concerns about their functional advantages relative to other Tfh subpopulations, Tfh1 and Tfh2, were raised by the reviewers, the authors nicely responded to their concerns. This study contributes to understanding the heterogeneity of memory Tfh cells and provides clues for better vaccine designs.

Follicular helper T (Tfh) cells are the specialized CD4$^+$ T cell subset that localize within B cell follicle to assist germinal center (GC) formation, plasma cell differentiation and high-affinity antibody production (*Vinuesa et al., 2016*; *Crotty, 2011*). Identified in the circulation and lymphoid organs post immune response (memory phase), CXCR5$^+$ memory Tfh cells rapidly differentiate into mature effector Tfh cells and accelerate antibody response upon antigen re-exposure (*Hale et al., 2013*; *He et al., 2013*; *MacLeod et al., 2011*; *Sage et al., 2014*; *Weber et al., 2012*; *Yu et al., 2022*).

CXCR5-expressing CD45RA$^-$ CD4$^+$ T cells circulating in human blood (cTfh) provide important subjects to investigate memory Tfh cells since they have egressed from the site of the immune response at secondary lymphoid organs and can differentiate into effector Tfh cells upon antigen re-exposure (*Tsai and Yu, 2014*). Noticeably, cTfh cells are heterogenous and are often classified into subsets by distinct functional markers. For example, cTfh cells are classified into cTfh1 (CXCR3$^+$CCR6$^-$), cTfh2 (CXCR3$^-$CCR6$^-$), and cTfh17 (CXCR3$^-$CCR6$^+$) subsets based on the expression of chemokine receptors CXCR3 and CCR6 associated with Th1 and Th17, respectively. cTfh2 and cTfh17 cells were reported to demonstrate better B cell helper function than cTfh1 cells in culture (*Morita et al., 2011*). The increases in cTfh2 and cTfh17 frequencies in autoimmune diseases often correlated with excessive production of pathogenic autoantibodies (*Zhao et al., 2018*; *Akiyama et al., 2015*). On the other hand, infections such as HIV and malaria mainly induce the generation of cTfh1 cells (*Niessl et al., 2020*; *Obeng-Adjei et al., 2015*). In influenza vaccination, it is also the cTfh1 subset that correlates with the titers of protective antibodies (*Bentebibel et al., 2013*).

Besides the classification of cTfh into cTfh1, cTfh2, and cTfh17 subsets by the features of lineage polarization, cTfh cells are composed of CCR7$^{high}$PD-1$^{low}$ 'central memory (CM)-like' (cTfh$_{CM}$) and CCR7$^{low}$PD-1$^{high}$ 'effector memory (EM)-like' (cTfh$_{EM}$) subsets (*He et al., 2013*), the latter also containing a more active ICOS$^+$ population (*Bentebibel et al., 2013*; *Spensieri et al., 2013*; *Herati et al., 2017*). CCR7$^{high}$ PD-1$^{low}$ cTfh$_{CM}$ cells are dominant in human blood cTfh cells whereas circulating CCR7$^{low}$PD-1$^{high}$ cTfh$_{EM}$ cells are temporarily induced in immune response and generated from the precursor stage of Tfh differentiation at secondary lymphoid organs (*He et al., 2013*).

There is little knowledge on the difference of Tfh1, Tfh2, and Tfh17 cells in memory responses. In this study, we developed a method to induce antigen-specific Tfh1, Tfh2, and Tfh17-like (iTfh1, iTfh2, and iTfh17) mouse cells in vitro. iTfh1, iTfh2, and iTfh17 cells showed comparable B-helper function after the adoptive transfer into recipient mice followed by immediate immunization. In contrast, if transferred cells experienced an extended period of resting before re-immunization, iTfh17 cells were superior to iTfh1 and iTfh2 cells in sustaining antibody responses. In humans, cTfh17 cells represented ~20% cTfh$_{EM}$ cells but accounted for >50% cTfh$_{CM}$ cells which transcriptionally and phenotypically resemble central memory CD4$^+$ T (T$_{CM}$) with better survival and proliferative potential than effector memory (T$_{EM}$) cells (*Sallusto et al., 2004*). In vaccine responses to hepatitis B virus (HBV), influenza, tetanus toxin or measles, the cTfh17 subset in vaccine-specific cTfh cells was preferentially maintained into memory phase and long-lived. Complementary results from mouse and human studies

**Table 1.** Summary of the Tfh populations in this study.

| Name | Species | Origin | Description |
| --- | --- | --- | --- |
| Tfh1 | Human, mouse | Lymphoid tissues, blood, in vitro culture | General nomenclature, refers to Th1-featured CXCR5-expressing CD4$^+$ T cells from all origins |
| Tfh2 | Human, mouse | Lymphoid tissues, blood, in vitro culture | General nomenclature, refers to Th2-featured CXCR5-expressing CD4$^+$ T cells from all origins |
| Tfh17 | Human, mouse | Lymphoid tissues, blood, in vitro culture | General nomenclature, refers to Th17-featured CXCR5-expressing CD4$^+$ T cells from all origins |
| Tfh | Human, mouse | Lymphoid tissues | Effector cells in the B cell follicle (CD4$^+$ CXCR5$^+$ PD-1$^+$) |
| GC-Tfh | Human, mouse | Lymphoid tissues | Effector cells in the germinal centre (CD4$^+$ CXCR5$^{high}$ PD-1$^{high}$BCL6$^{high}$) |
| iTfh1 | Mouse | In vitro culture | Culture induced Tfh1-like cells (CD44$^+$ PD-1$^+$ CXCR5$^+$ BCL6$^+$ T-bet$^+$) |
| iTfh2 | Mouse | In vitro culture | Culture induced Tfh2-like cells (CD44$^+$ PD-1$^+$ CXCR5$^+$ BCL6$^+$ GATA3 $^+$) |
| iTfh17 | Mouse | In vitro culture | Culture induced Tfh17-like cells (CD44$^+$ PD-1$^+$ CXCR5$^+$ BCL6$^+$ RORγt $^+$) |
| cTfh | Human | Blood | Circulating memory Tfh cells (CD4$^+$ CD45RA$^-$ CXCR5$^+$) |
| cTfh$_{CM}$ | Human | Blood | Circulating memory Tfh cells with central memory features (CD4$^+$ CXCR5$^+$ CCR7$^{high}$ PD-1$^{low}$) |
| cTfh$_{EM}$ | Human | Blood | Circulating memory Tfh cells with effector memory features (CD4$^+$ CXCR5$^+$ CCR7$^{low}$ PD-1$^{high}$) |
| cTfh1 | Human | Blood | Circulating memory Tfh1 cells (CD4$^+$ CD45RA$^-$ CXCR5$^+$ CXCR3$^+$ CCR6$^-$) |
| cTfh2 | Human | Blood | Circulating memory Tfh2 cells (CD4$^+$ CD45RA$^-$ CXCR5$^+$ CXCR3$^-$ CCR6$^-$) |
| cTfh17 | Human | Blood | Circulating memory Tfh17 cells (CD4$^+$ CD45RA$^-$ CXCR5$^+$ CXCR3$^-$ CCR6$^+$) |

thus suggest Tfh17 cells are superior for memory maintenance, the ability to persist and to support humoral response upon antigen restimulation. Of note, this study includes many Tfh populations and their definitions and features were summarized to facilitate clarification (*Table 1*).

# Results
## In vitro differentiation of induced Tfh1, Tfh2, and Tfh17-like (iTfh1, iTfh2, iTfh17) cells

The classification of Tfh1/2/17 cells based on the expression of CXCR3 and CCR6 markers was established by characterizing human blood memory Tfh cells (*Morita et al., 2011*). Such memory Tfh1/2/17 subsets in mice are low in numbers (*Figure 1A*), thus limiting functional characterization. We modified an established method that induces antigen-specific naive CD4$^+$ T cells, such as OT-II T cells with transgenic TCR specific to ovalbumin (OVA), to differentiate into Tfh cells (iTfh) in vitro (*Gao et al., 2020*) and induced the individual differentiation into Tfh1/2/17 (iTfh1/2/17) in vitro. In addition to IL-6 and IL-21 in the iTfh differentiation method (*Gao et al., 2020*), Th1 (IL-12, anti-TGF-β, anti-IL-4), Th17 (TGF-β, anti-IFN-γ, anti-IL-4), and Th2 (IL-4, anti-IFN-γ, anti-TGF-β) polarization cultures were adopted

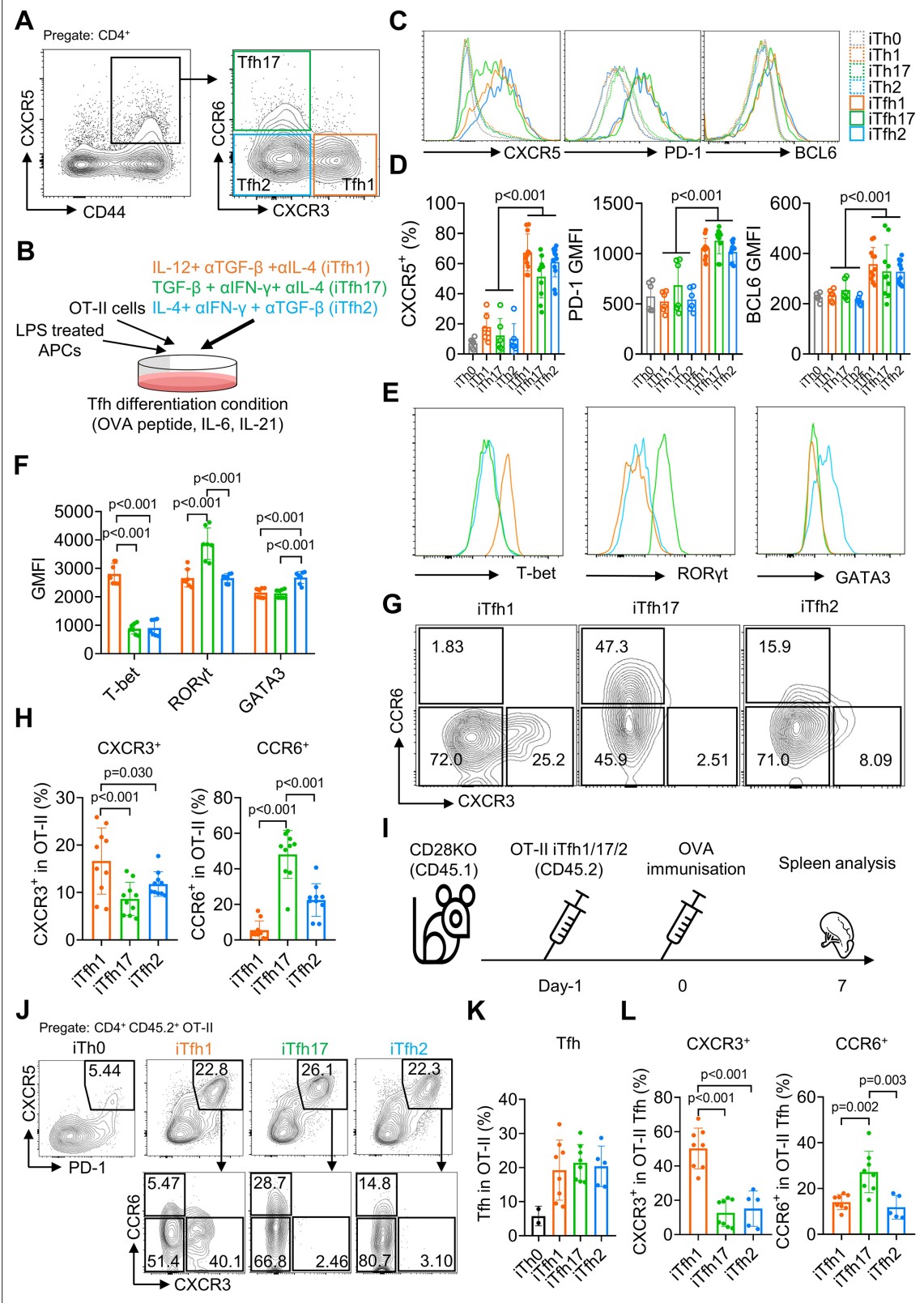

**Figure 1.** The in vitro differentiation of induced Tfh1, Tfh2 and Tfh17-like (iTfh1, iTfh2, iTfh17) cells. (**A**) Splenocytes from WT mice were analyzed and representative FACS plot for Tfh1, Tfh17, and Tfh2 cells was shown. (**B–H**) OT-II cells were co-cultured with WT splenocytes as antigen-presenting cells (APCs) in the presence of OVA peptide, indicated cytokines and blocking antibodies for three days before phenotypic analysis. Experiment design (**B**), representative FACS plots for the expression of Tfh markers CXCR5, PD-1 and BCL6 (**C**) transcription factors T-bet, RORγt and GATA3 (**E**), CXCR3 *vs*

*Figure 1 continued on next page*

*Figure 1 continued*

CCR6 expression (**G**) and statistics (**D, F, H**). (**I–L**) $5 \times 10^4$ cultured OT-II iTh0, iTfh1, iTfh2, and iTfh17 cells were FACS-purified and separately transferred into CD28KO recipients, followed by OVA-Alum immunization. The spleens were collected on day7 post-immunization for FACS analysis. Experiment design (**I**), representative FACS plot for Tfh percentage in OT-II cells (**J**), statistics of Tfh percentage in OT-II cells (**K**) and statistics of CXCR3/CCR6[+] percentage in OT-II Tfh cells (**L**). The p values were calculated by two-way ANOVA for (**D**) and one-way ANOVA for (**F, H, L**). The results in (**D, F, H**) were pooled from three independent experiments. The results in (**K, L**) were pooled from two independent experiments. Source data for the statistics can be found in *Figure 1—source data 1*.

The online version of this article includes the following source data and figure supplement(s) for figure 1:

**Source data 1.** Source data file of statistics in *Figure 1*.

**Figure supplement 1.** iTfh1/2/17 cells show lower BCL6 expression than GC-Tfh cells.

**Figure supplement 1—source data 1.** Source data file of statistics in *Figure 1—figure supplement 1*.

for iTfh1/2/17 induction with lower IL-12, TGF-β or IL-4 concentrations for iTfh1/2/17 induction than those used for canonical iTh1, iTh17, or iTh2 induction (details in the method) (*Read et al., 2019*; *Lu et al., 2011*; *Nurieva et al., 2008*; *Figure 1B*). iTfh1/2/17 cells expressed higher Tfh-defining markers CXCR5, PD-1 and BCL6 than iTh0/1/2/17 cells (*Figure 1C and D*). The BCL6 expression in iTfh1/2/17 cells was lower than CD44+CXCR5highPD-1high GC-Tfh cells in immunized mice (*Figure 1—figure supplement 1A, B*). Tfh differentiation undergoes a step-by-step process, showing the generation of precursor Tfh cells expressing intermediate levels of BCL6 and subsequent maturation of GC-Tfh cells with the highest BCL6 expression (*Vinuesa et al., 2016*; *Crotty, 2011*). Memory cTfh cells largely originate from precursor Tfh cells and express low levels of BCL6 (*He et al., 2013*). Given that iTfh1/2/17 cells expressed BCL6 lower than that in GC-Tfh cells and resemble precursor Tfh cells, iTfh1/2/17 cells are suitable to study the function of memory cTfh cells. Importantly, iTfh1/2/17 cells differentially expressed transcription factors T-bet, GATA3 and RORγt (*Figure 1E and F*) and chemokine receptors CXCR3 and CCR6 (*Figure 1G and H*), as their counterpart human Tfh1/2/17 cells (*Morita et al., 2011*).

To examine whether iTfh1/2/17 cells retain polarized phenotypes in vivo, we adoptively transferred each cell type individually into congenic CD28KO recipient mice, followed by the immunization of OVA in aluminium salt (OVA-Alum) (*Figure 1I*). After 7 days, iTfh1/2/17 cells showed the comparable ability of effector Tfh differentiation (*Figure 1J and K*) but maintain the distinction in CXCR3 and CCR6 expression aligning with their progenitors (*Figure 1J and L*). These results suggest that in vitro generated iTfh1/2/17 cells can be used to investigate the function of Tfh1/2/17 cells.

## iTfh17 cells are superior in memory maintenance

To compare the function of Tfh1, Tfh2 and Tfh17 in vivo, we adoptively transferred each of OT-II naive T cell-derived iTfh1, iTfh2, or iTfh17 cells into congenic CD28KO recipient mice. T cells in CD28KO mice are defective in co-stimulation and unable to generate endogenous Tfh cells so antibody responses in CD28KO mice are dependent on transferred iTfh cells. After adoptive cell transfer, mice were immunized with OVA-Alum at day 0 (early immunization) or day 35 (late immunization) with the latter condition mimicking memory maintenance of Tfh1, Tfh2, and Tfh17 for extended in vivo resting (*Figure 2A*).

On day 7 post the early immunization, OT-II iTfh1, iTfh2 and iTfh17 cells demonstrated largely comparable Tfh differentiation and the function in supporting the generation of germinal centre B (B$_{GC}$) cells and antibody-secreting B (B$_{ASC}$) cells (*Figure 2B–G*). A larger magnitude of B$_{GC}$ cells supported by iTfh2 cells might be explained by the function of IL-4 in enhancing B$_{GC}$ generation (*Gaya et al., 2018*). However, the outcome was very different in the scheme of late immunization whereby iTfh cells had experienced memory maintenance. After resting in vivo for 35 days, iTfh17 cells produced more than twofold of mature effector Tfh cells than those by iTfh1 or iTfh2 cells (*Figure 2B and E*), accompanied by ~twofold increase in B$_{GC}$ differentiation in mice that had received iTfh17 cells than those had received iTfh1 or iTfh2 cells (*Figure 2C and F*). Although the trend of an increase in iTfh17-supported B$_{ASC}$ differentiation didn't reach statistical significance (*Figure 2D and G*), iTfh17 cells were superior to iTfh1 and iTfh2 cells in supporting the production of anti-NP IgG antibodies, after resting in vivo for 35 days before the immunization with antigen 4-Hydroxy-3-nitrophenyl (NP)-OVA (*Figure 2H*).

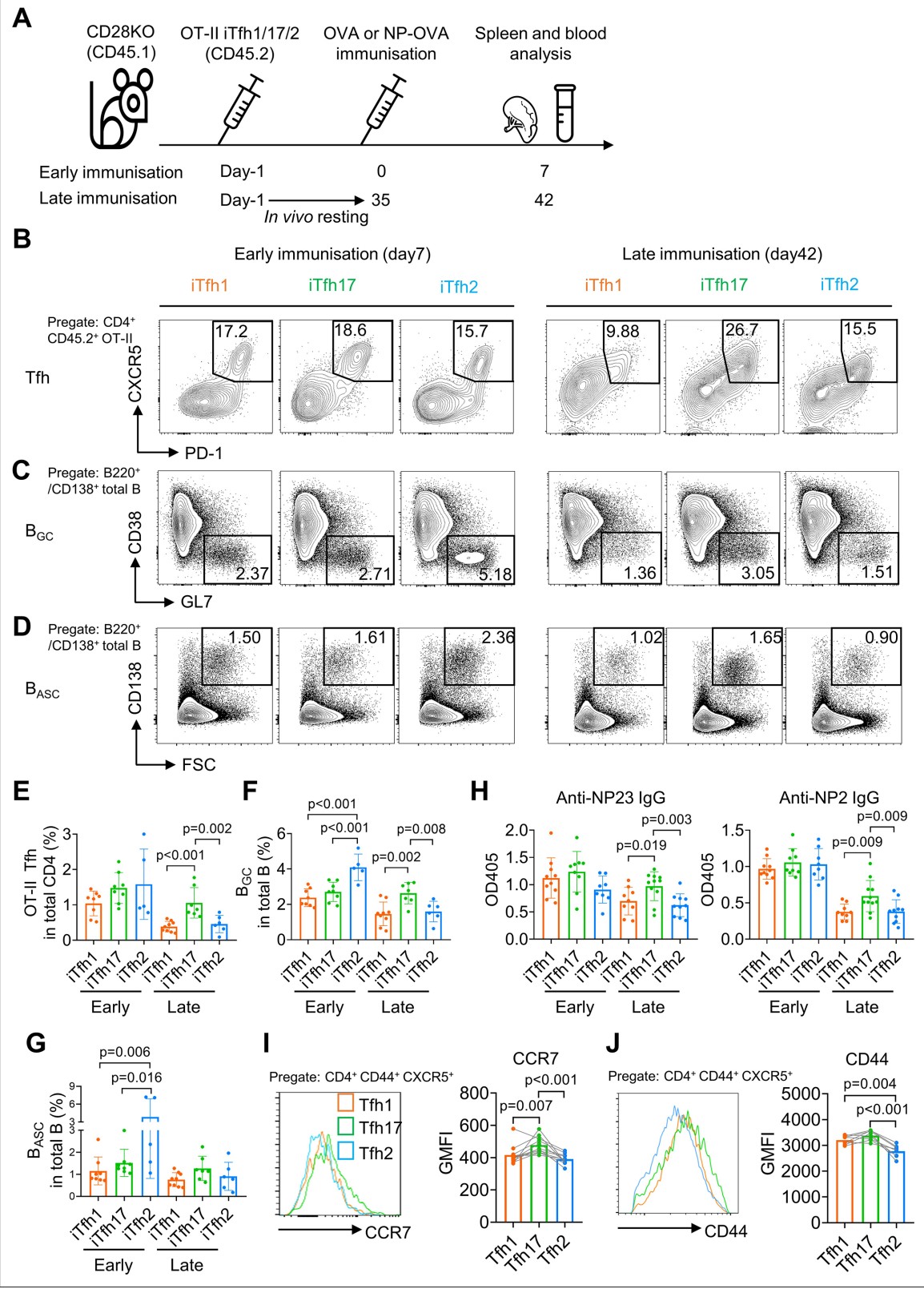

**Figure 2.** iTfh17 cells are superior in memory maintenance. (**A–H**) 5×10⁴ FACS-purified OT-II iTfh1, iTfh17, or iTfh2 cells were separately transferred to CD28KO recipients. The early immunization group was immunized by OVA or NP-OVA in alum one day after the adoptive cell transfer. The late immunization group was immunized by the same antigens 35 days after the adoptive cell transfer. Spleens or serum were collected on day 7 after the immunization. Experiment design (**A**), representative FACS plots (**B, C, D**) and statistics (**E, F, G**) showing the percentages of Tfh cells in OT-II cells, the

*Figure 2 continued on next page*

*Figure 2 continued*

percentages of $B_{GC}$ in total B cells and the percentages of $B_{ASC}$ in total B cells. For antibody titers, statistic (**H**) showing OD405 values of anti-NP2 and anti-NP23 total IgG. (**I–J**) Tfh1/2/17 cells from mouse splenocytes were analyzed for CCR7 and CD44 expression. Representative FACS plots (**I**). and statistics (**J**) showing the expressions of CCR7 and CD44. The p values were calculated by one-way ANOVA. The results in (**E, F, G, H, I, J**) were both pooled from two independent experiments. Source data for the statistics can be found in *Figure 2—source data 1*.

The online version of this article includes the following source data for figure 2:

**Source data 1.** Source data file of statistics in *Figure 2*.

Collectively, iTfh17 cells are superior to iTfh1 or iTfh2 cells in helping B cells, but only in the scheme with extended in vivo resting.

The selective advantage of iTfh17 cells in supporting Tfh differentiation and humoral immunity after an extended in vivo resting followed by immunization suggests that Tfh17 cells may outperform Tfh1 or Tfh2 cells to sustain Tfh memory. In resting mice, Tfh17 cells expressed higher CCR7 than that on Tfh1 or Tfh2 cells (*Figures 1A and 2I*), despite the highest expression of activation marker CD44 by Tfh17 cells (*Figure 2J*). CCR7$^+$ marks $T_{CM}$ cells that circulate in the blood and secondary lymphoid tissues and have longer survival and better proliferative capacity than CCR7$^-$ $T_{EM}$ cells (*Sallusto et al., 2004*; *Bouneaud et al., 2005*). We thus hypothesized that Tfh17 cells might carry certain features of $T_{CM}$ cells suitable for memory maintenance.

## Human cTfh$_{CM}$ and cTfh$_{EM}$ subsets phenotypically and functionally resemble $T_{CM}$ and $T_{EM}$ subsets respectively

Following the observation that iTfh17 cells showed a unique advantage in memory maintenance, we set to characterize the function of human Tfh17 cells in maintaining Tfh memory. We previously reported that human cTfh cells are composed of CCR7$^{high}$PD-1$^{low}$ $T_{CM}$-like and CCR7$^{low}$PD-1$^{high}$ $T_{EM}$-like subsets with the latter indicating an active Tfh differentiation (*He et al., 2013*), but their function has not been formally compared. We first investigated the relationship between the two cTfh subsets and corresponding CD4$^+$ $T_{CM}$ and $T_{EM}$ subsets by transcriptomic analysis using RNA sequencing (RNA-seq) (*Figure 3A*). As shown in an unsupervised multidimensional scaling (MDS) plot, cTfh$_{CM}$ cells closely cluster with $T_{CM}$ cells, and cTfh$_{EM}$ cells fall between $T_{CM}$ and $T_{EM}$ cells on the major dimension1, implying that cTfh$_{EM}$ cells are distinct from cTfh$_{CM}$ and $T_{CM}$ cells but also have effector programs different from $T_{EM}$ cells (*Figure 3B*). Previous studies reported that cTfh cells predominantly show CCR7$^+$ CM phenotype (*Chevalier et al., 2011*). Our transcriptomic analysis indeed suggests that cTfh$_{CM}$ cells acquire a quiescent state hardly distinguishable from $T_{CM}$ cells. In contrast, cTfh$_{EM}$ cells' transcriptomes are clearly separated from those of $T_{EM}$ cells, presumably caused by the divergent effector function of Tfh cells as compared to other effector Th1, Th2, or Th17 cells. In line with this, the top 50 differentially expressed genes (DEG) indicate effector genes such as *ZEB2* and *TBX21* (*Omilusik et al., 2015*) were highly expressed in $T_{EM}$ cells, intermediate levels in cTfh$_{EM}$ cells, and lowest in cTfh$_{CM}$ and $T_{CM}$ cells (*Figure 3C*). In top 50 hallmark gene sets identified by gene set enrichment analysis (GSEA) between cTfh$_{EM}$ *vs* cTfh$_{CM}$ cells or $T_{EM}$ *vs* $T_{CM}$ cells, 37 gene sets were significantly enriched by both comparisons (NES discrepancy >2, *Figure 3D*), suggesting that the transcriptomic features and regulation between cTfh$_{EM}$ *vs* cTfh$_{CM}$ cells are overall similar to those between $T_{EM}$ *vs* $T_{CM}$. Despite $T_{EM}$ and cTfh$_{Em}$ cells show distinct transcriptomes and locate separately in the MDS plot (*Figure 3B*), the key gene sets that are related to common effector T cell function (activation, effector differentiation, and cell cycle entry) were both positively enriched in comparisons between cTfh$_{EM}$ *vs* cTfh$_{CM}$ cells or $T_{EM}$ *vs* $T_{CM}$ cells (*Figure 3E*). Therefore, cTfh$_{CM}$ and cTfh$_{EM}$ cells resemble $T_{CM}$ and $T_{EM}$ cells at the transcriptomic levels respectively.

We next compared cTfh$_{CM}$ and cTfh$_{EM}$ subsets for survival and stimulation-induced proliferation in culture, which were applied to characterize the difference between $T_{CM}$ and $T_{EM}$ cells (*Sallusto et al., 2004*). In non-stimulation culture for 3 days, $T_{CM}$ and cTfh$_{CM}$ cells retained ~50% viability while $T_{EM}$ and cTfh$_{EM}$ cells showed poorer survival of ~30% (*Figure 3F and G*). To measure the proliferative potential, all subsets were labeled with carboxyfluorescein succinimidyl ester (CFSE) and stimulated by anti-CD3/CD28 for 2.5 days. While the majority of $T_{EM}$ or cTfh$_{EM}$ cells underwent division once, most $T_{CM}$ or cTfh$_{CM}$ cells reached the second or third division, indicating a better proliferative potential (*Figure 3H, I*). Collectively, CCR7$^{high}$PD-1$^{low}$ cTfh$_{CM}$ and CCR7$^{low}$PD-1$^{high}$ cTfh$_{EM}$ subsets showed not

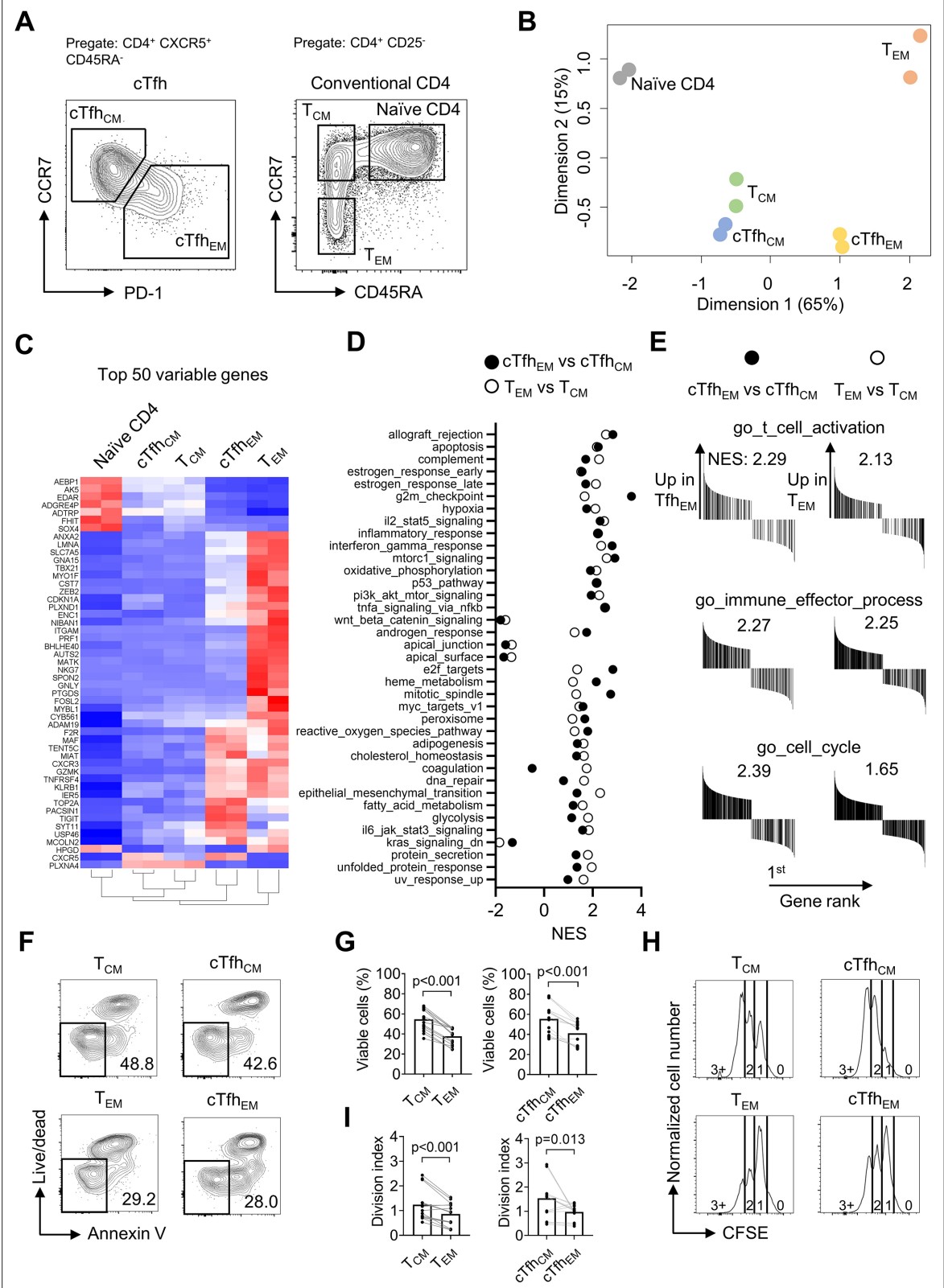

**Figure 3.** Human cTfh_CM and cTfh_EM subsets phenotypically and functionally resemble T_CM and T_EM subsets respectively. (**A–E**) Naïve, T_CM, T_EM, cTfh_CM , and cTfh_EM cells were FACS-purified from PBMC of two healthy donors and bulk RNA-seq was performed for differentially expressed genes analysis and gene set enrichment analysis (GSEA). (**A**) Representative FACS plot showing the gating strategy for indicated subsets. (**B**) MDS plot showing sample distribution. (**C**) Heatmap of the top 50 variable genes normalized by z-score. (**D**) Summarized normalized enrichment score (NES) of significantly

*Figure 3 continued on next page*

*Figure 3 continued*

enriched (p<0.05, FDR <0.25) hallmark gene sets by either cTfh$_{EM}$ vs cTfh$_{CM}$ or T$_{EM}$ vs T$_{CM}$. (**E**) GSEA on selected gene sets were performed on cTfh$_{EM}$vs cTfh$_{CM}$ and T$_{EM}$ vs T$_{CM}$ and the number indicates NES. (**F–G**) FACS-purified T$_{CM}$, T$_{EM}$, cTfh$_{CM}$ and cTfh$_{EM}$ cells were rested in complete RPMI for 3 days. Representative FACS plots (**F**) and statistics (**G**) showing the percentages of viable cells. (**H–I**) FACS-purified T$_{CM}$, T$_{EM}$, cTfh$_{CM}$, and cTfh$_{EM}$ cells were labelled with CFSE and stimulated by anti-CD3/CD28 for 2.5 days. Representative FACS plots (**H**) and statistics (**I**) showing the CFSE fluorescence intensity and the division index. The p values were calculated by Wilcoxon matched-pairs signed-rank test. The results in (**G, I**) were pooled from ive healthy individuals with each conducted in three technical replicates. Source data for the statistics can be found in *Figure 3—source data 1*.

The online version of this article includes the following source data for figure 3:

**Source data 1.** Source data file of statistics in *Figure 3*.

only transcriptomic profiles resembling their counterpart T$_{CM}$ and T$_{EM}$ cells but also functional characteristics of survival and proliferative capacity (*Sallusto et al., 2004*; *Bouneaud et al., 2005*).

## cTfh$_{CM}$ cells are enriched with the cTfh17 subset whereas cTfh$_{EM}$ cells are enriched with the cTfh1 subset

From a cohort of healthy donors (N=33, *Table 2*), we analyzed CCR7$^{high}$PD-1$^{low}$ cTfh$_{CM}$ and CCR7$^{low}$PD-1$^{high}$ cTfh$_{EM}$ cells for the percentages of cTfh1/2/17 subsets based on CXCR3 and CCR6 expression. In agreement with the higher expression of CCR7 on mouse Tfh17 than that on Tfh1 or Tfh2 cells (*Figure 2I*), human cTfh$_{CM}$ cells were dominated by the cTfh17 subset (mean = 51.44%), followed by the cTfh2 subset (mean = 16.19%) and the cTfh1 subset (mean = 12.17%) (*Figure 4A*), whereas cTfh$_{EM}$ cells were dominated by the cTfh1 subset (mean = 34.06%,) (*Figure 4B*). The population of cTfh cells that expresses both CXCR3 and CCR6 has been reported and also presented in our samples. Due to the fact that CXCR3$^+$CCR6$^+$ cTfh cells were fewer than cTfh1, cTfh2, or cTfh17 cells and their ontogeny remains to be fully revealed (*Morita et al., 2011*), we did not include this population in the following analyses. To avoid the influence of individual variation of Tfh1/2/17 polarization due to different histories of immune exposure and examine whether there is an intrinsic difference of cTfh1/2/17 frequencies between cTfh$_{CM}$ or cTfh$_{EM}$ cells, the percentages of cTfh1/2/17 cells in cTfh$_{CM}$ cells were normalized to those in cTfh$_{EM}$ cells in each individual, which demonstrated the highest cTfh$_{CM}$/cTfh$_{EM}$ ratio for cTfh17 (mean = 4.18), an intermediate ratio for cTfh2 (mean = 3.38), and the lowest ratio for cTfh1 (mean = 0.75) (*Figure 4C*). The highest ratio (>>1) for cTfh17 indicates cTfh$_{CM}$ cells are highly enriched with the cTfh17 subset whereas the lowest ratio (<1) for cTfh1 indicates that cTfh$_{EM}$ cells are enriched with the

**Table 2.** Demographics for all human samples included in the research.

| Cohort description | Number | Gender (female, male) | Age (median, range) | Corresponding Figures |
|---|---|---|---|---|
| Healthy individuals for cTfh phenotyping | 33 | 26/7 | 35 (21–71) | *Figure 4A–C* *Figure 7D–E* *Figure 5—figure supplement 1A–C* |
| Healthy individuals received HBV vaccines | 38 | 8/29 | 19 (18–20) | *Figure 5* *Figure 7D–E* *Figure 5—figure supplement 1D–E* *Figure 5—figure supplement 2* |
| Healthy individuals for measles and TT AIM assay | 20 | 11/9 | 24 (18–32) | *Figure 7D–E* |
| Healthy children | 18 | 14/4 | 6 (0.5–12) | *Figure 7D–E* |
| Cord blood | 5 | 2/3 | 0 (0–0) | *Figure 7D–E* |
| Recovered Covid-19 patients | 13 | 9/4 | 33 (23–52) | *Figure 7—figure supplement 1A–C* |
| Healthy individuals for qPCR and cytokine assay | 14 | 9/5 | 42.5 (27-51) | *Figure 4D–G* *Figure 7—figure supplement 1A* |

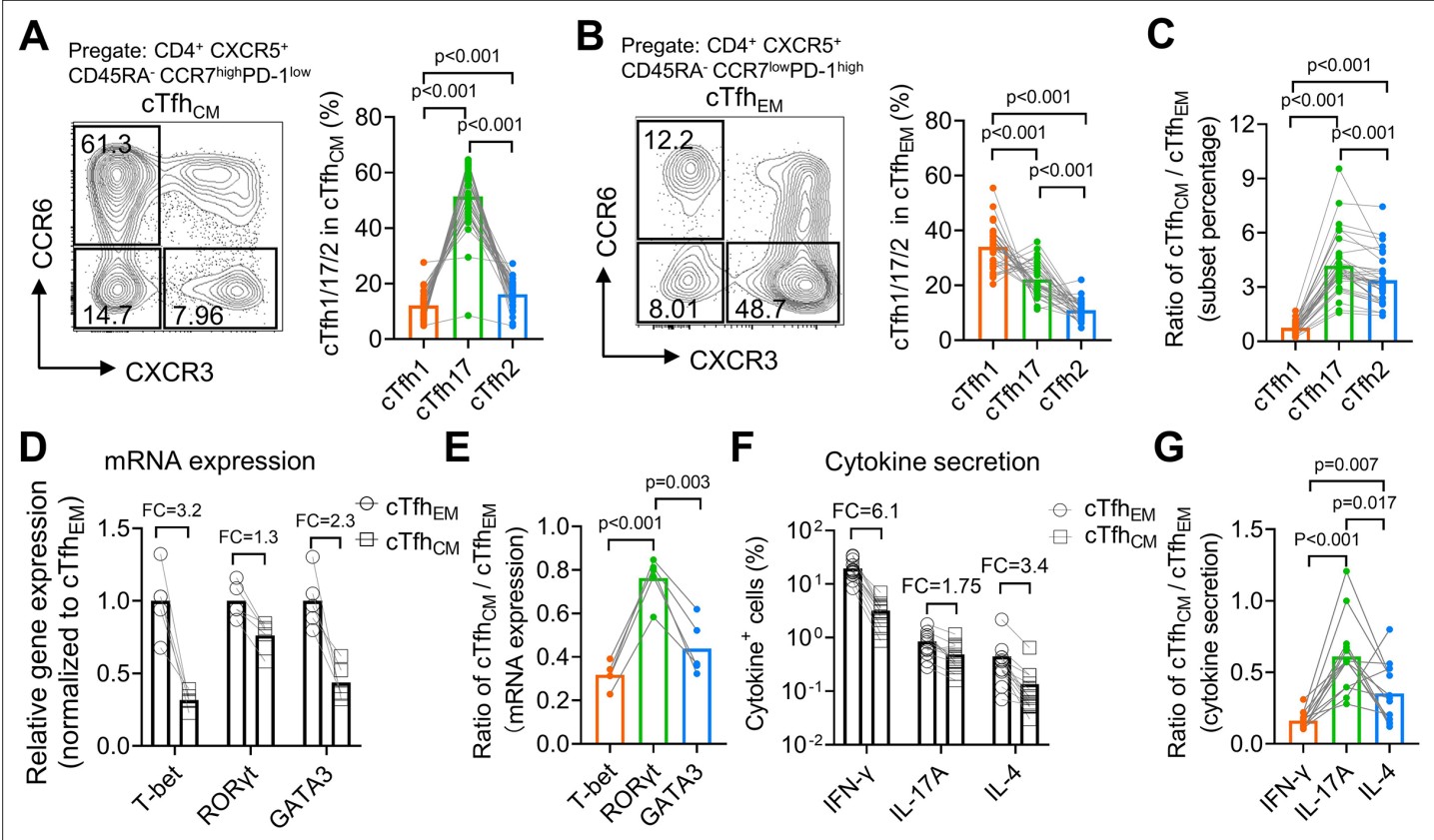

**Figure 4.** Human cTfh_CM cells are enriched with the cTfh17 subset whereas cTfh_EM cells are enriched with the cTfh1 subset. (A–C) Human PBMC samples from 33 healthy blood donors were analyzed. Representative FACS plots and statistics showing the percentages of cTfh1, cTfh2, and cTfh17 cells in cTfh_CM (A) or cTfh_EM (B) subsets. cTfh_CM/cTfh_EM ratios for cTfh1/2/17 in each individual were calculated (C). (D–E) FACS-purified cTfh_EM and cTfh_CM from five healthy individuals were analyzed for the expressions of indicated transcription factors by qPCR. The statistics for relative gene expression $2^{-\Delta\Delta Ct}$ (normalized to cTfh_EM) (D) and cTfh_CM/cTfh_EM ratios (E). (F–G) PBMC from 13 healthy individuals were analyzed for the secretions for indicated cytokines post PMA/ionomycin stimulation. The statistics for the percentages of cytokine+ cells (F) and the cTfh_CM/cTfh_EM ratios (G). FC: average fold change. The $p$ values were calculated by Friedman test. Source data for the statistics can be found in *Figure 4—source data 1*.

The online version of this article includes the following source data for figure 4:

**Source data 1.** Source data file of statistics in *Figure 4*.

cTfh1 subset. We also measured the expression of hallmark transcription factors *TBX21*, *GATA3*, and *RORC* and cytokines IFN-γ, IL-4 and IL-17A that are selectively expressed in cTfh1/2/17 cells, respectively (*Morita et al., 2011*). These molecules for effector Th functions are abundantly expressed by T_EM cells but are downregulated in T_CM cells which enter into a resting state (*Sallusto et al., 2004*). Indeed, the expression of effector transcription factors and cytokines was consistently lower in cTfh_CM cells than those in cTfh_EM cells (*Figure 4D and F*). Notably, the ratios of expression (cTfh_CM/cTfh_EM) demonstrate modest reductions of 20–40% in cTfh17-related markers RORγt and IL-17A, in contrast to vast reductions of 70–80% in cTfh1-related markers T-bet and IFN-γ (*Figure 4E and G*). Such results of transcription factor and cytokine expression support the conclusion for an enrichment of the cTfh17 subset and a loss of the cTfh1 subset in cTfh_CM cells.

## HBV antigen-specific cTfh17 cells are preferentially maintained in memory phase

cTfh_EM to cTfh_CM phenotype conversion occurs over the period of a few weeks when the antigen stimulation is discontinued (*He et al., 2013*). The enrichment of the cTfh17 subset in human cTfh_CM cells suggest the cTfh17 subset in cTfh_EM cells may persist longer than the cTfh1 or cTfh2 subsets. The phenomenon could also result from a biased cTfh_CM phenotype of cTfh17 cells generated even early in immune responses. To tease apart the cause, we next examined the phenotype of antigen-specific

human cTfh cells over a period that expands both effector and memory phases after vaccination or infection.

Childhood HBV vaccination doesn't always provide life-long protection with a proportion of vaccinees with antibody titers at an undetectable level in adulthood (*Bruce et al., 2016*). HBV boosting vaccination is recommended for high-risk populations such as medical practitioners including medical students in China. A cohort of medical students (N=38) with serum negative for anti-HBV surface antigen (HBVSA) antibody were recruited (*Table 2*). Peripheral blood mononuclear cells (PBMCs) were collected at day 7 before and day 7 and 28 after the immunization (*Figure 5A*). Antigen-induced marker (AIM) assay was used to examine HBV vaccine-specific T cells by culturing PBMCs with HBVSA for 18 hr and detecting the PD-L1$^+$OX40$^+$CD25$^+$ cells as the antigen-specific population (*Figure 5B*). This method has been applied to characterize antigen-specific cTfh cells (*Dan et al., 2016*; *Reiss et al., 2017*). Such stimulation did not change the expression of CXCR3 and CCR6 on cTfh cells (*Figure 5— figure supplement 1A–C*), indicating that AIM assays are suitable to characterize antigen-specific cTfh1/2/17 cells. As reported (*Reiss et al., 2017*), a background in AIM assays exists in a small proportion of samples whereby PD-L1$^+$OX40$^+$CD25$^+$ cells were detected in control cultures without antigen stimulation (*Figure 5—figure supplement 1D*). To specifically quantify antigen-specific response, we subtracted the value of antigen-stimulation culture by the background value from the control culture without antigen stimulation. Normalized values were then used to calculate the percentages of cTfh1, cTfh2, and cTfh17 cells (*Figure 5—figure supplement 1E*).

In all subjects negative for HBVSA antigen and anti-HBVSA antibody, the average percentage of the cTfh17 subset in HBVSA-specific memory cTfh cells was 49.31%, whereas the average percentage of the cTfh1 subset was 15.7% (*Figure 5C and D*). In alignment with the results on total cTfh$_{CM}$ cells (*Figure 4*), HBVSA-specific cTfh$_{CM}$ cells were also enriched with the cTfh17 subset.

To tease apart whether the cTfh17 enrichment was caused by the biased generation or better maintenance, we then analysed PBMC samples collected at day 7 and day 28 after vaccination (*Figure 5A*). The cohort was divided into 4 groups based on vaccine responses measured by antibody titers (*Figure 5—figure supplement 2A*): early responders (titer >100 mIU at day7, titer = 1248 ± 324.4 mIU at day 28, N=11), late responders (titer <100 mIU at day 7 and >200 mIU at day 28, titer = 997.1 ± 289.7 mIU/ml at day 28, N=15), weak responders (50 mIU <titer < 200 mIU at day 28, titer = 107.9 ± 37.2 mIU at day 28, N=6) and non-responders (titer <50 mIU on day 28, titer = 17.28 ± 4.379 mIU at day 28, N=6) (*Figure 5E*). Tfh activation measured by a trend of increase in HBVSA-specific cTfh$_{EM}$ cells by (2.84-fold, day –7 v.s. day 7, -value=0.074, not reaching statistical significance) was observed only in early responders but no other groups (*Figure 5—figure supplement 2B, C*). We next focused on the kinetics of HBVSA-specific cTfh subsets in early responders. At day 7 post vaccination, the percentages of three subsets in HBVSA-specific cTfh cells ranged from 20% to 30% and showed no significant difference (*Figure 5—figure supplement 2D*). Of note, the percentages of the cTfh17 subset significantly increased from ~30% to~50% from day 7–28 (p-value = 0.006). In contrast, the percentages of the cTfh1 subset dropped significantly from ~20% to~10% (p-value = 0.020). The percentages of cTfh2 remained largely unchanged (*Figure 5F and G*). As a result, cTfh17 cells dominated HBVSA-specific cTfh cells in the memory phase of day 28 post vaccination (*Figure 5—figure supplement 2D*). Therefore, the cTfh17 enrichment in cTfh$_{CM}$ cells results from an advantage of cTfh17 cells in memory maintenance, rather than a biased phenotype to cTfh$_{CM}$ cells. Significant changes in cTfh1 and cTfh17 percentages from day 7 to day 28 were selective in early responder group but not in three other groups (*Figure 5G*), and only observed in HBVSA-specific cTfh cells but not in total cTfh in early responders (*Figure 5—figure supplement 2E*), suggesting that the dynamic changes were specific to HBVSA-specific cTfh response.

## influenza virus-specific cTfh cells show cTfh1 signatures in effector phase but cTfh17 signatures in memory phase

Single-cell RNA-seq (scRNA-seq) paired with TCR sequencing facilitates the characterization of the phenotype and function of antigen-specific T cell clones in an immune response. We took the advantage of this new technology to analyze the characteristics of influenza haemagglutinin (HA)-specific CD4$^+$ T clones in a published dataset of scRNA/TCR-seq from four healthy individuals with influenza vaccination (*Meckiff et al., 2020*). We compared HA-specific T cell clones before the vaccination (memory phase) and day 12 after the vaccination (effector phase; *Figure 6A*).

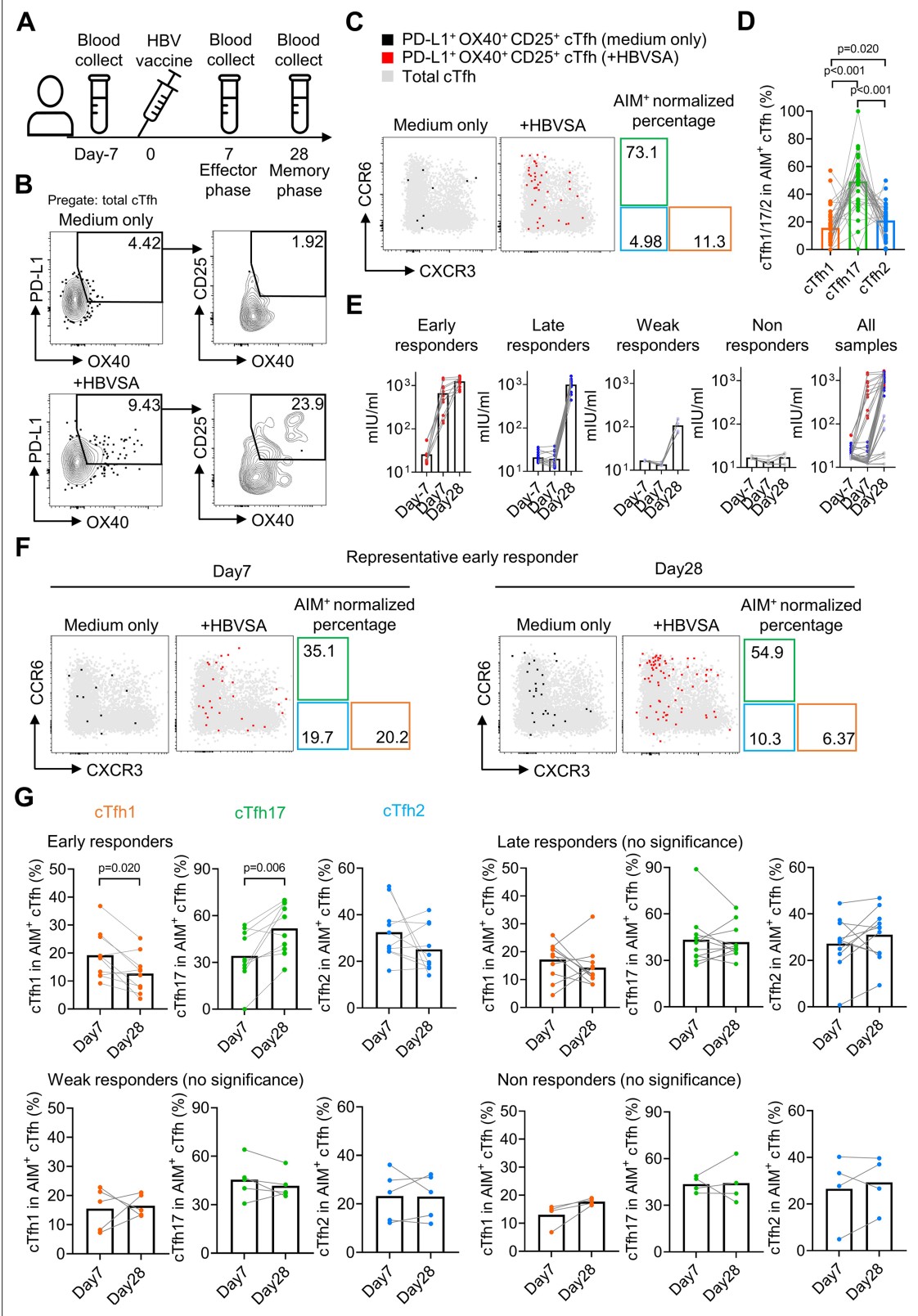

**Figure 5.** HBV antigen-specific cTfh17 cells are preferentially maintained in memory phase. Blood samples from HBV vaccinated healthy individuals (N=38) were collected on indicated time points before/after HBV vaccination, and serum was diluted 10 times to analyse the anti-HBVSA antibody titer by ELISA. PBMC were also isolated and cultured with or without 20 μg/mL HBVSA for 18 hr, followed by FACS to analyse the phenotype of HBVSA-specific cTfh cells. Experiment design (**A**) and representative FACS plot (**B**) showing the gating strategy to detect HBVSA-specific cTfh cells by AIM

*Figure 5 continued on next page*

*Figure 5 continued*

assay. Representative FACS plot (**C**) and statistics (**D**) showing the percentage of cTfh1/2/17 cells in HBVSA-specific cTfh cells before vaccination. Classification (**E**) of 38 individuals into four groups was based on their anti-HBVSA antibody titers. Representative FACS plot (**F**) for an early responder showing the percentage of cTfh1/2/17 cells in HBVSA-specific cTfh. Statistics (**G**) showing the percentage of cTfh1/2/17 cells in HBVSA-specific cTfh on day 7 and day 28 after the vaccination in all defined groups (N=30, 8 samples with poor signals in AIM assay were excluded). The p values were calculated by Wilcoxon matched-pairs signed-rank test. Source data for the statistics can be found in *Figure 5—source data 1*.

The online version of this article includes the following source data and figure supplement(s) for figure 5:

**Source data 1.** Source data file of statistics in *Figure 5*.

**Figure supplement 1.** AIM assays to identify antigen-specific cTfh cells.

**Figure supplement 1—source data 1.** Source data file of statistics in *Figure 5—figure supplement 1*.

**Figure supplement 2.** The cTfh responses in HBV vaccinated individuals.

**Figure supplement 2—source data 1.** Source data file of statistics in *Figure 5—figure supplement 2*.

HA-specific CD4$^+$ T cells before and after the vaccination were pooled to generate unsupervised clustering, in which CXCR5-expressing clusters 2–5 were enriched of cTfh cells, in which a total of twelve major CD4$^+$ T clones (clonal abundance ≥10) were identified (*Figure 6B*). To investigate Tfh subsets-associated features in HA-specific clonal cTfh cells, we applied cTfh1 or cTfh17 signature gene sets derived from bulk RNA-seq for cTfh1/2/17 cells (*Figure 6—figure supplement 1A*; *Yost et al., 2019*) in clonal cTfh cells. The scores of cTfh1 signature were higher in clonal cTfh cells in the effector phase than those in the memory phase (p-value = 0.041); by contrast, the scores of cTfh17 signature were lower in the effector phase than in the memory phase (p-value = 0.002) (*Figure 6C and D*). The divergence of cTfh1 and cTfh17 signatures between the effector and memory phases was consistent in individual donors (*Figure 6—figure supplement 1B*) and at the level measured by 12 major clones (*Figure 6E and F*). Therefore, we conclude that the advantage of cTfh17 cells in memory maintenance is consistently observed among different cohorts with different types of vaccines.

## cTfh17 cells are long-lived and accumulate with aging

Our previous experiments have demonstrated that antigen-specific mouse Tfh17 cells and vaccine-specific human Tfh17 cells are superior to Tfh1 and Tfh2 cells in maintaining Tfh memory for a period of about one month (HBV) or less than one year (influenza vaccine). We next ask whether Tfh17 cells can persist for even longer periods, such as years. In a cohort of adults (N=20, *Table 2*), we examined cTfh cells specific to vaccines for tetanus toxoid and measles, both administrated in childhood (*Figure 7A*). Given that community transmission of tetanus and measles is very rare (*Huang et al., 2018*), cTfh cells specific to tetanus toxoid and measles in adults were likely induced many years ago by childhood vaccination (*Van Damme et al., 2019*; *Nanan et al., 2000*; *Locci et al., 2013*). The average percentages of the cTfh17 subset in vaccine-specific memory cTfh cells were 55.22% and 45.07% for tetanus toxoid and measles respectively, which were more than twofold higher than the cTfh2 percentages and more than threefold higher than the cTfh1 percentages (*Figure 7B and C*). These results suggest cTfh17 cells may maintain Tfh memory for more than a decade. We also asked whether the cTfh17 dominance in memory phase was also applied to cTfh cells induced by SARS-CoV-2 infection. We examined convalescent patients with Covid-19 showing SARS-CoV-2-specific IgG antibodies (N=13, *Figure 7—figure supplement 1A* and *Table 2*). Similar to vaccine-specific cTfh cells, the cTfh17 percentages in SARS-CoV-2-specific cTfh cells were much higher than the cTfh1 or cTfh2 percentages (mean, 59.03% v.s. 12.87% or 7.73%) (*Figure 7—figure supplement 1B, C*).

If antigen-specific cTfh17 cells are long-lived and superior to cTfh1 and cTfh2 cells for persistence, we would expect a preferential accumulation of cTfh17 cells over cTfh1 or cTfh2 cells along with aging. By pooling results of cTfh characterization from cord blood, children, young and middle-aged adults and the elderly (*Table 2*), we indeed observed that the cTfh17 percentages in total cTfh cells positively correlated with biological ages whereas the percentages of cTfh1 or cTfh2 subsets showed negative correlations (p-value <0.001) (*Figure 7D and E*). In conclusion, Tfh17 cells are superior to Tfh1 and Tfh2 cells in Tfh memory maintenance, a phenomenon consistently observed in vaccination, infection and natural antigen exposure.

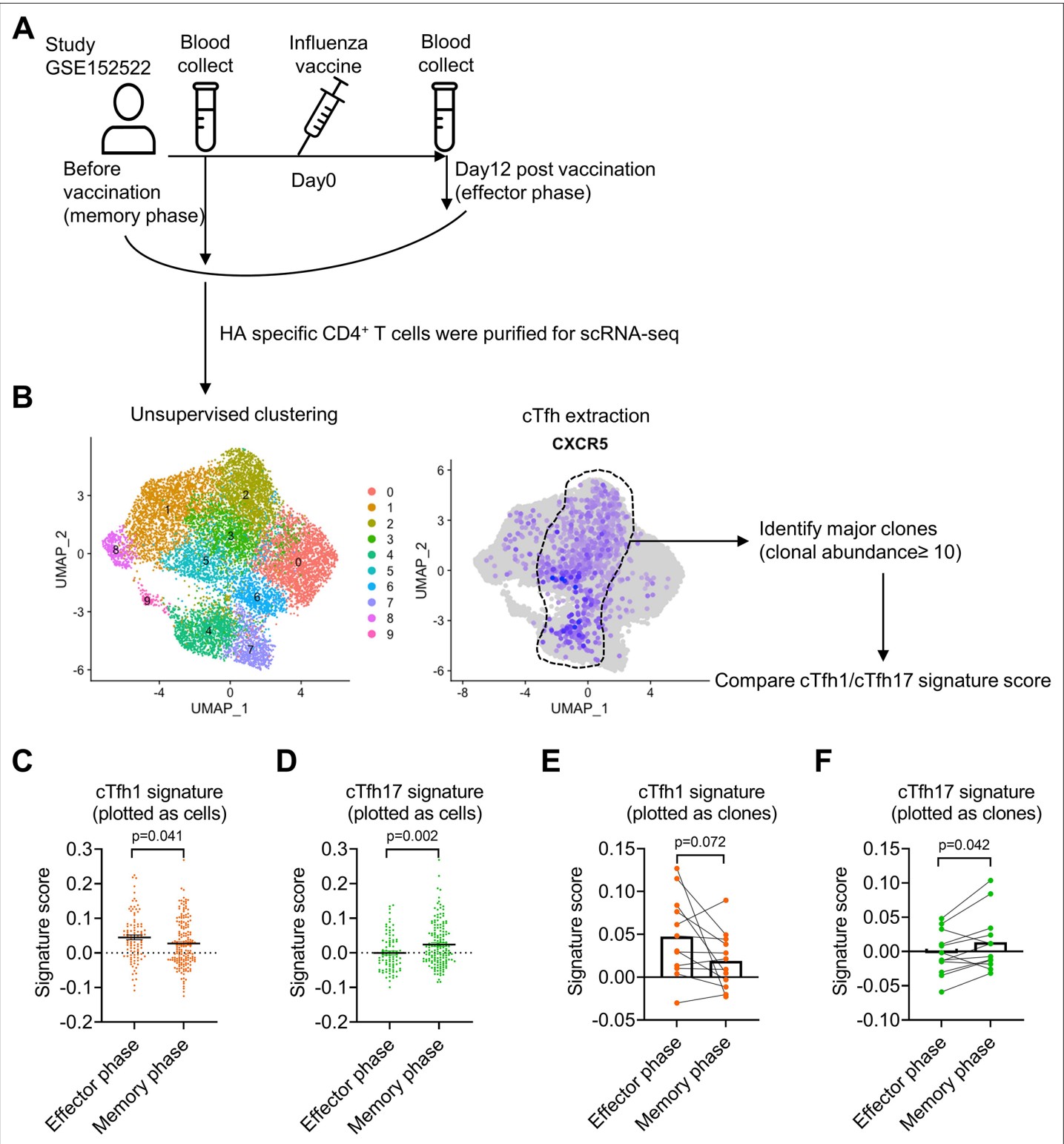

**Figure 6.** Influenza virus-specific cTfh cells show cTfh1 signatures in effector phase but cTfh17 signatures in memory phase. The single-cell RNA-seq dataset (GSE152522, the experiment design **A**) was analyzed to identify CXCR5-expressing cTfh clusters (**B**), which contain 12 major clones with a total of 249 cells. Comparison of cTfh1 and cTfh17 signature scores between effector and memory phase cTfh cells based on each cell or clone were shown in (**C, D**) and (**E, F**). The signature score of each clone was calculated as the mean value of the signature scores of all the cells in this clone. The *p* values were calculated by unpaired *t*-tests for (**C, D**) and paired *t*-tests for (**E, F**). Source data for the statistics can be found in *Figure 6—source data 1*.

The online version of this article includes the following source data and figure supplement(s) for figure 6:

*Figure 6 continued on next page*

*Figure 6 continued*

**Source data 1.** Source data file of statistics in *Figure 6*.

**Figure supplement 1.** scRNA-seq analysis for influenza-specific cTfh cells.

**Figure supplement 1—source data 1.** Source data file of statistics in *Figure 6—figure supplement 1*.

## iTfh17 cells are superior in survival and differentiation into GC-Tfh cells after resting

We reasoned that the advantage of Tfh17 cells in supporting humoral responses after delayed immunization might be attributed to several non-exclusive reasons: (1) Tfh17 can survive better; (2) Tfh17 cells can maintain stronger potential to differentiate into GC-Tfh cells after resting, and (3) Tfh17-derived GC-Tfh cells can gain better B cell helper function. Firstly, to test whether Tfh17 cells can better survive than Tfh1/2 cells, we transferred either OT-II iTfh1, iTfh2 or iTfh17 cells into CD28KO mice and counted the numbers of transferred cells in the spleen after 1 day and 35 days (*Figure 8A*). While the numbers of transferred iTfh1/2/17 cells were comparable on day1, the numbers of transferred iTfh17 cells were significantly higher than iTfh1 cells on day35 (*Figure 8B–C*), suggesting that iTfh17 cells had superior survival capacity over iTfh1 but not iTfh2 cells.

Secondly, to test whether Tfh17 cells may maintain better potential to differentiate into GC-Tfh cells after resting, we transferred either OT-II iTfh1, iTfh2 or iTfh17 cells into CD28KO mice together with NP-specific B1-8 cells, followed by an immediate NP-OVA immunization at day 1 or a delayed NP-OVA immunization to examine the formation of GC-Tfh cells (*Figure 8D*). In the immediate immunization, iTfh1/2/17 cells expanded and differentiated into GC-Tfh in comparable manners after immunization (*Figure 8E–G*). However, in the delayed immunization (day 35), iTfh17 cells showed higher expansion than iTfh1 but not iTfh2 cells (*Figure 8E*). Furthermore, iTfh17 cells differentiated into more GC-Tfh cells than both iTfh1 and iTfh2 cells (*Figure 8F–G*). These results suggest that iTfh17 cells maintained a better potential to generated GC-Tfh cells compared to Tfh1 or Tfh2 cells, in addition to a better survival than iTfh1 cells.

Finally, to compare the B cell helper function between iTfh1/2/17-derived GC-Tfh cells on per cell basis, we sorted iTfh1/2/17-derived CXCR5$^{hi}$ PD-1$^{hi}$ GC-Tfh cells in the same experiment as in '(2)' and measured the expressions of key functional genes in GC-Tfh cells including *Pdcd1, Cxcr5, Icos, Cd40lg, Il21,* and *Bcl6*. In the delayed immunization, we found no significant differences in these gene expression among iTfh1/2/17-derived GC-Tfh cells, despite of better B$_{GC}$ and B$_{ASC}$ responses in the iTfh17 group (*Figure 8—figure supplement 1A–B*). In summary, our results suggested that the superior immunological memory maintenance of iTfh17 cells was attributed to their better survival capacity and better maintenance of the potential to differentiate into GC-Tfh cells, rather than better B cell helper function on per cell basis than that of iTfh1 or iTfh2 cells.

Furthermore, we measured the expressions of *Ifng* and *Il4* in iTfh1/2/17 derived GC-Tfh cells and demonstrated iTfh1 and iTfh2-derived GC-Tfh cells showed featured of increased *Ifng* and *Il4* respectively, as their counterpart Th1 and Th2 cells (*Figure 8—figure supplement 1C*). In agreement with polarized cytokine profiles, we detected that iTfh1 cells promoted isotype switching to IgG2a/IgG3 while iTfh2 cells promoted isotype switching to IgG1/IgE (*Figure 8—figure supplement 1D*). These results suggest iTfh1/2 cells retained polarised cytokine profiles that promote specific class-switch recombination after antigen re-exposure.

## Discussion

Morita et al. reported circulating Tfh memory cells comprise different subsets related to Th1, Th2, and Th17 cells (*Morita et al., 2011*), which has advanced our understanding of Tfh subsets and helped to delineate the relationship between Tfh and other Th subsets. While the signatures of Tfh2 and Tfh17 activation were commonly reported in allergic and autoimmune diseases (*Deng et al., 2019*; *Yao et al., 2021a*), the activation of Tfh1 cells was a prominent feature and associated with pathogen-specific antibody production in influenza vaccination (*Bentebibel et al., 2013*) and infections by HIV (*Baiyegunhi et al., 2018*), malaria (*Obeng-Adjei et al., 2015*) and more recently SARS-CoV-2 (*Rydznski Moderbacher et al., 2020*; *Dan et al., 2021*). Beyond their distinct function in mediating isotype class switching (*Gowthaman et al., 2019*; *Hirota et al., 2013*), other difference

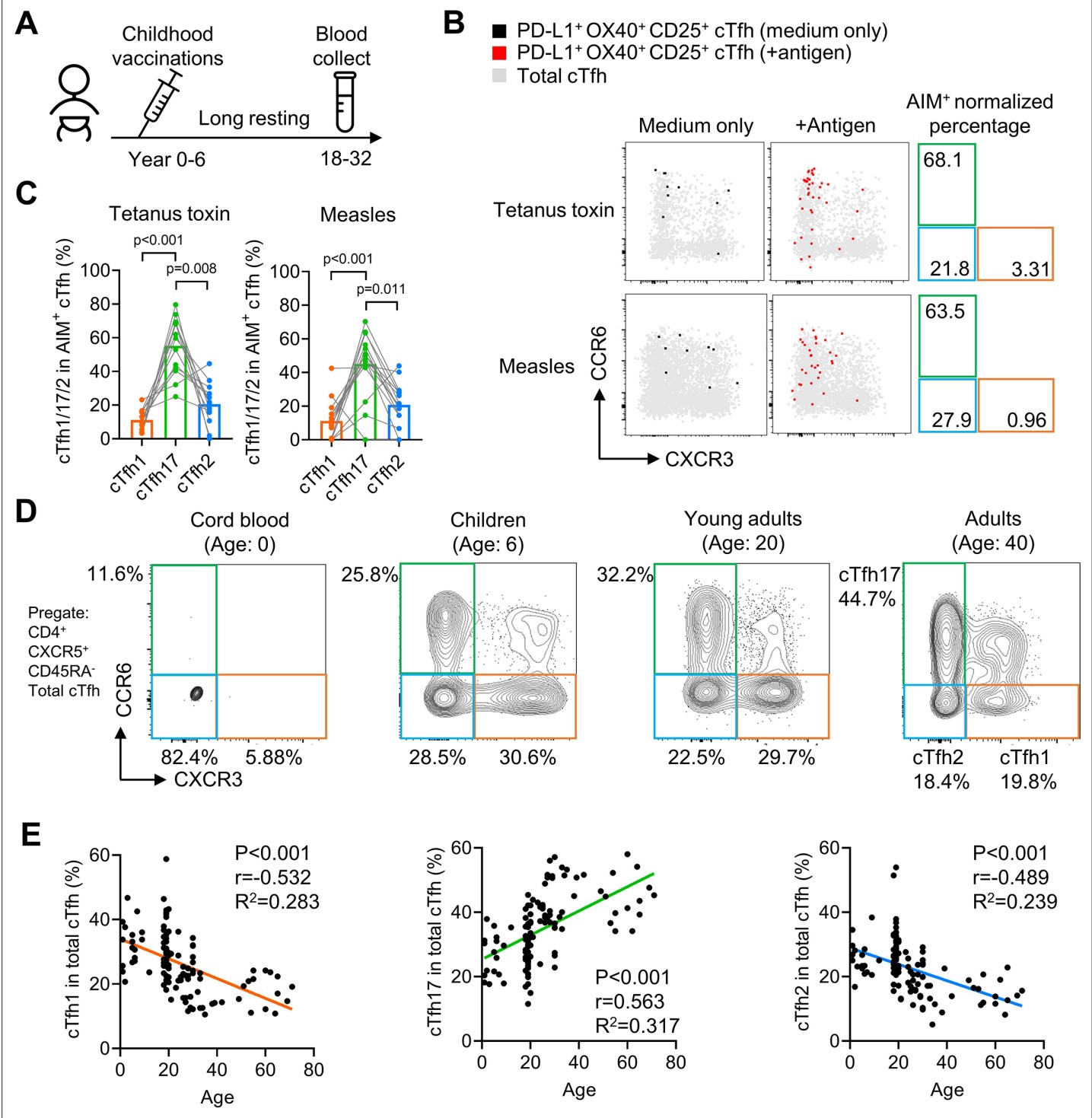

**Figure 7.** cTfh17 cells are long-lived and accumulate with aging. (**A–C**) PBMC samples from 20 healthy individuals were cultured for 18 hr with or without indicated antigens, followed by FACS to detect the phenotype of antigen-specific cTfh cells. Experiment design (**A**), representative FACS plot (**B**) and statistics (**C**) showing the percentage of cTfh1/2/17 cells in antigen-specific cTfh cells against tetanus toxin or measles. (**D–F**) PBMC samples from individuals of different ages were analysed. Representative FACS plots (**D**) showing the percentages of cTfh1/2/17 cells in total cTfh cells in individuals of different ages. Correlations tests (**E**) between the biological age with the percentages of cTfh1/2/17 cells in total cTfh cells. Cord blood samples were excluded from the correlation tests because of insufficient cTfh cell numbers. The p values were calculated by Friedman test for (**C**) and Pearson correlation for (**E**). Source data for the statistics can be found in *Figure 7—source data 1*.

The online version of this article includes the following source data and figure supplement(s) for figure 7:

*Figure 7 continued on next page*

*Figure 7 continued*

**Source data 1.** Source data file of statistics in *Figure 7*.

**Figure supplement 1.** SARS-CoV-2-specific cTfh17 cells have superior persistence.

**Figure supplement 1—source data 1.** Source data file of statistics in *Figure 7—figure supplement 1*.

between these Tfh subsets remains largely unknown, possibly due to the experimental hurdle of a low frequency of Tfh subsets in human blood and the lack of culture method for in vitro Tfh subset generation and in vivo functional characterization.

We adopted the published method for the in vitro induction of antigen-specific Tfh differentiation (*Gao et al., 2020*) and modified the protocol by adding the conditions biased for Th1/2/17 polarization. This method corroborated the report that Tfh cells are plastic and carry positive epigenetic markings for Th1/2/17 cells (*Lu et al., 2011*). Notably, for the iTfh17 condition (0.1 ng/mL TGF-β+100 ng/mL IL-6 + 50 ng/mL IL-21), TGF-β were used in a concentration of 0.1 ng/mL, much lower than those for Th17 and Treg polarization (normally 1–10 ng/mL). The condition successfully generated iTfh17 expressing both CXCR5, PD-1, BCL6 and RORγt. This phenomenon might also help to reconcile the reports that TGF-β signaling can either inhibit or support Tfh differentiation (*McCarron and Marie, 2014*; *Marshall et al., 2015*), probably determined by the TGF-β signal strength.

The generation of antigen-specific iTfh1/2/17 cells in decent numbers using this modified method allowed us, for the first time, to compare Tfh1, Tfh2 and Tfh17 function in vivo. The results from the adoptive transfer experiment revealed that, after an extended period of in vivo resting to mimic memory maintenance, iTfh17 cells showed a better function than iTfh1/2 cells in supporting humoral immunity. In agreement with this, the cTfh17 subset in human cTfh cells also showed superiority over cTfh1 and cTfh2 subsets for memory maintenance. cTfh17 cells predominantly showed the cTfh$_{CM}$ phenotype as long-lived memory cells and dominated the long-lived pool of antigen-specific memory Tfh cells for vaccines of HBV, influenza, tetanus and measles. In contrast, the human cTfh1 subset appears short-lived and accounted for the least proportion of long-lived antigen-specific memory cTfh cells. Tfh1 is the major Tfh subset induced by influenza infection and vaccination (*Bentebibel et al., 2013*). The short-lived characteristics of Tfh1 cells may partially contribute to a relatively short period of humoral immunity after influenza vaccination (*Young et al., 2018*). In SARS-CoV-2 infection, patients with acute infection demonstrated a Tfh1-biased profile, while convalescent patients increased the proportions of virus-specific Tfh17 cells, again supporting the notion that Tfh17 cells represent the population better in memory maintenance (*Rydyznski Moderbacher et al., 2020*; *Dan et al., 2021*). In malaria infection, a recent study also reported Tfh17 and Tfh2 cells, rather than Tfh1 cells showed a Tfh$_{CM}$ phenotype (*Chan et al., 2020*). All such evidence suggests the superiority of Tfh17 subset in memory maintenance appears to be a common feature for immune responses induced by both vaccination and infection. It should be noted that our results should not be misinterpreted as that Tfh17 cells are always the major subset for Tfh memory cells. In the case of SARS-CoV-2 mRNA vaccine which induces strong Th1-polarized response, Tfh17 cells are essentially not induced and the Tfh memory are maintained in the absence of Tfh17 cells (*Goel et al., 2021*; *Wragg et al., 2022*; *Tauzin et al., 2021*).

Our results suggest that the superior immunological memory maintenance of iTfh17 cells was attributed to their better survival capacity and better maintenance of the potential to differentiate into GC-Tfh cells. The molecular mechanism underlying the advantage of Tfh17 cells in immunological memory maintenance is an area we will focus on in the following studies. Intriguingly, Th17 cells were reported to have 'stem cell-like' features and are long-lived (*Lindenstrøm et al., 2012*; *Kryczek et al., 2011*; *Muranski et al., 2011*). The proposed mechanisms are diverse, which include a high expression of Tcf1 in Th17 cells, a key transcription factor that regulates T cell memory generation and self-renewal and favorable expression of anti-apoptosis Bcl-2 family genes to sustain the longevity (*Kryczek et al., 2011*; *Muranski et al., 2011*). The Th17's hallmark transcription factor RORγ has been shown to directly promote T cell survival by enhancing Bcl-xL expression (*Sun et al., 2000*). Effector and memory Tfh cells are critically regulated by specific cell death pathways of ferroptosis and pyroptosis (*Yao et al., 2021b*; *Chen et al., 2022*). Future works are required to test whether these mechanisms also apply in the survival and self-renewal of Tfh17 cells.

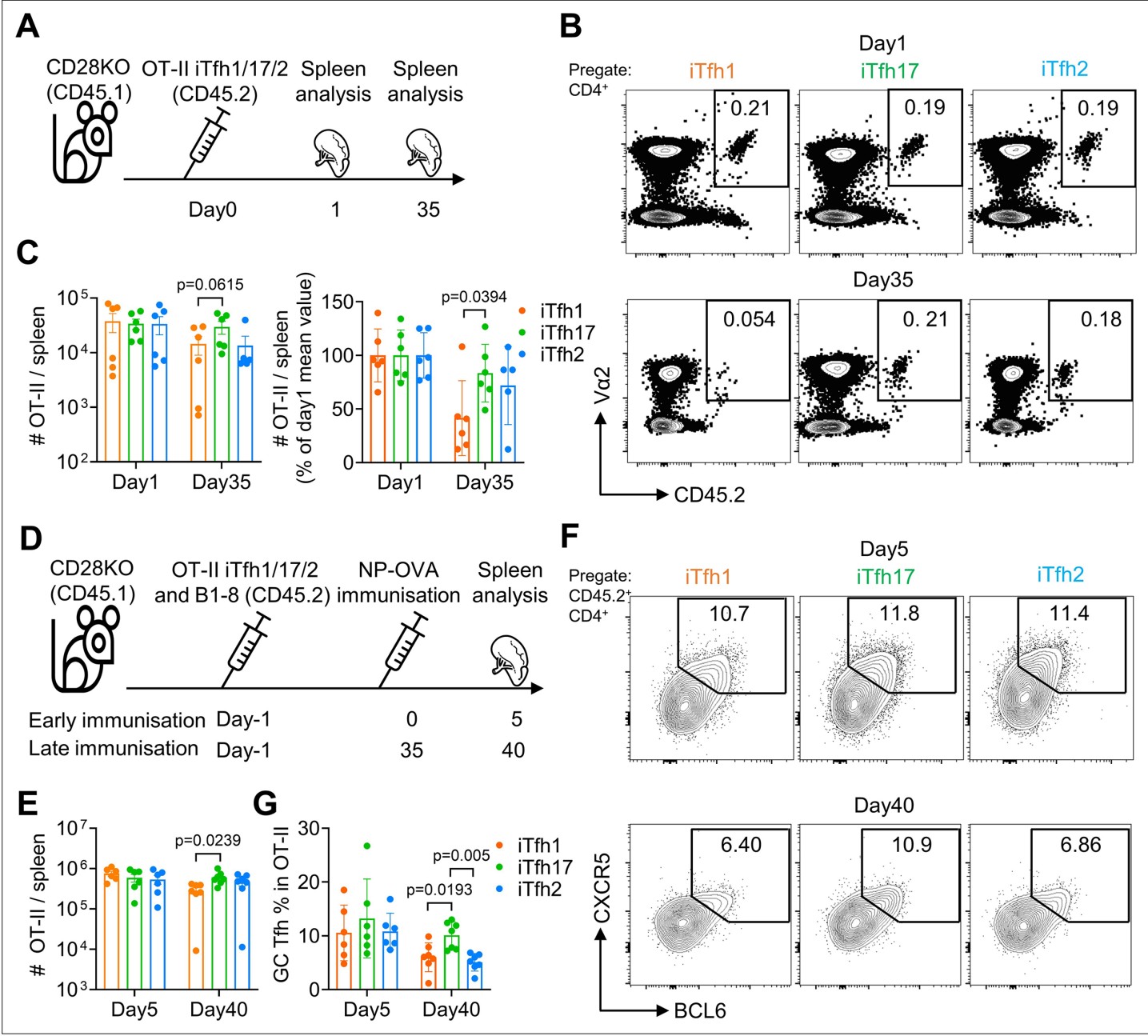

**Figure 8.** iTfh17 cells are superior in survival and differentiation into GC-Tfh cells after resting. (**A–C**) 5×10⁴ FACS-purified OT-II iTfh1, iTfh17, or iTfh2 cells were separately transferred to CD28KO recipients, and the spleens were FACS analysed on day1 and day35. Experimental design (**A**), representative FACS plots (**B**) and statistics (**C**) showing the total and normalized numbers of transferred iTfh1/2/17 cells in the spleens. (**D–G**) FACS-purified 1×10⁴ B1-8 B cells and 5×10⁴ OT-II iTfh1, iTfh17 or iTfh2 cells were co-transferred to CD28KO recipients. The early immunization group was immunized by NP-OVA in alum 1 day after the adoptive cell transfer. The late immunization group was immunized by the same antigens 35 days after the adoptive cell transfer. Spleens were collected on day 5 after the immunization. Experiment design (**D**). Statistic showing the numbers of OT-II cells in the spleen (**E**). Representative FACS plots (**F**) and statistics (**G**) showing the percentages of GC Tfh cells in OT-II cells. The p values were calculated by one-way ANOVA. The results in (**C, E, G**) were both pooled from two independent experiments. Source data for the statistics can be found in *Figure 8—source data 1*.

The online version of this article includes the following source data and figure supplement(s) for figure 8:

**Source data 1.** Source data file of statistics in *Figure 8*.

**Figure supplement 1.** iTfh1/2 cells retain polarized Th1/2 cytokine profiles and promote specific isotype class switching.

**Figure supplement 1—source data 1.** Source data file of statistics in *Figure 8—figure supplement 1*.

Unveiling the superiority of Tfh17 in Tfh memory maintenance can help us to improve the rationale-based vaccine development. Many vaccines, including conventional vaccines for influenza virus and novel mRNA vaccines for SARS-CoV-2, induce Th1 responses and Tfh1-associated humoral immunity (*Bentebibel et al., 2013*; *Sahin et al., 2020*; *Lederer et al., 2020*). According to our results, Tfh1 cells are short-lived, which might curb the duration of vaccine-mediated protection. New strategies might be taken to direct vaccination for more Tfh17 induction which can support a better Tfh memory formation and potentially prolong vaccine protection.

# Materials and methods

**Key resources table**

| Reagent type (species) or resource | Designation | Source or reference | Identifiers | Additional information |
|---|---|---|---|---|
| Sequence-based reagent | *TBX21-F* | IDT | PCR primers | CACTACAGGATGTTTGTGGACGTG |
| Sequence-based reagent | *TBX21-R* | IDT | PCR primers | CCCCTTGTTGTTTGTGAGCTTTAG |
| Sequence-based reagent | *GATA3-F* | IDT | PCR primers | TGTCTGCAGCCAGGAGAGC |
| Sequence-based reagent | *GATA3-R* | IDT | PCR primers | ATGCATCAAACAACTGTGGCCA |
| Sequence-based reagent | *RORC -F* | IDT | PCR primers | TCTGGAGCTGGCCTTTCATCATCA |
| Sequence-based reagent | *RORC -R* | IDT | PCR primers | TCTGCTCACTTCCAAAGAGCTGGT |
| Sequence-based reagent | *GAPDH -F* | IDT | PCR primers | TGCACCACCAACTGCTTAG |
| Sequence-based reagent | *GAPDH -R* | IDT | PCR primers | GGATGCAGGGATGATGTTC |
| Sequence-based reagent | *Pdcd1-F* | IDT | PCR primers | CGGTTTCAAGGCATGGTCATTGG |
| Sequence-based reagent | *Pdcd1-R* | IDT | PCR primers | TCAGAGTGTCGTCCTTGCTTCC |
| Sequence-based reagent | *Cxcr5-F* | IDT | PCR primers | ATCGTCCATGCTGTTCACGCCT |
| Sequence-based reagent | *Cxcr5-R* | IDT | PCR primers | CAACCTTGGCAAAGAGGAGTTCC |
| Sequence-based reagent | *Icos-F* | IDT | PCR primers | GCAGCTTTCGTTGTGGTACTCC |
| Sequence-based reagent | *Icos-R* | IDT | PCR primers | TGTGTTGACTGCCGCCATGAAC |
| Sequence-based reagent | *Cd40lg-F* | IDT | PCR primers | GAACTGTGAGCAGATGAGAAGGC |
| Sequence-based reagent | *Cd40lg-R* | IDT | PCR primers | TGGCTTCGCTTACAACGTGTGC |
| Sequence-based reagent | *Il21-F* | IDT | PCR primers | GCCTCCTGATTAGACTTCGTCAC |
| Sequence-based reagent | *Il21-R* | IDT | PCR primers | CAGGCAAAAGCTGCATGCTCAC |
| Sequence-based reagent | *Bcl6-F* | IDT | PCR primers | CAGAGATGTGCCTCCATACTGC |
| Sequence-based reagent | *Bcl6-R* | IDT | PCR primers | CTCCTCAGAGAAACGGCAGTCA |
| Sequence-based reagent | *Ifng-F* | IDT | PCR primers | CAGCAACAGCAAGGCGAAAAAGG |
| Sequence-based reagent | *Ifng-R* | IDT | PCR primers | TTTCCGCTTCCTGAGGCTGGAT |
| Sequence-based reagent | *Il4-F* | IDT | PCR primers | ATCATCGGCATTTTGAACGAGGTC |
| Sequence-based reagent | *Il4-R* | IDT | PCR primers | ACCTTGGAAGCCCTACAGACGA |
| Sequence-based reagent | *Il17a-F* | IDT | PCR primers | CAGACTACCTCAACCGTTCCAC |
| Sequence-based reagent | *Il17a-R* | IDT | PCR primers | TCCAGCTTTCCCTCCGCATTGA |
| Sequence-based reagent | Ubc-F | IDT | PCR primers | GCCCAGTGTTACCACCAAGA |
| Sequence-based reagent | *Ubc-R* | IDT | PCR primers | CCCATCACACCCAAGAACA |
| Antibody | Anti-human-CD4, mouse monoclonal | Biolegend | Clone: RPA-T4 | 1:200 |
| Antibody | Anti-human- CD45RA, mouse monoclonal | Biolegend | Clone: HI100 | 1:200 |

*Continued on next page*

*Continued*

| Reagent type (species) or resource | Designation | Source or reference | Identifiers | Additional information |
|---|---|---|---|---|
| Antibody | Anti-human- CXCR5, mouse monoclonal | Biolegend | Clone: J252D4 | 1:100 |
| Antibody | Anti-human- CXCR3, mouse monoclonal | Biolegend | Clone: G025H7 | 1:100 |
| Antibody | Anti-human- CCR6, mouse monoclonal | Biolegend | Clone: G034E3 | 1:50 |
| Antibody | Anti-human- CCR7, mouse monoclonal | Biolegend | Clone: G043H7 | 1:100 |
| Antibody | Anti-human- PD-1, mouse monoclonal | Biolegend | Clone: A17188B | 1:50 |
| Antibody | Anti-human- PD-L1, mouse monoclonal | Biolegend | Clone: 29E.2A3 | 1:30 |
| Antibody | Anti-human- OX40, mouse monoclonal | Biolegend | Clone: Ber-ACT35 (ACT35) | 1:200 |
| Antibody | Anti-human- CD25, mouse monoclonal | Biolegend | Clone: BC96 | 1:100 |
| Antibody | Anti-human- CD19, mouse monoclonal | Biolegend | Clone: HIB19 | 1:200 |
| Antibody | Anti-human- IFN-γ, mouse monoclonal | Biolegend | Clone: B27 | 1;100 |
| Antibody | Anti-human- IL-4, mouse monoclonal | Biolegend | Clone: MP4-25D2 | 1:50 |
| Antibody | Anti-human- IL-17A, mouse monoclonal | Biolegend | Clone: BL168 | 1:100 |
| Antibody | Anti-mouse- B220, rat monoclonal | Biolegend | Clone: RA3-6B2 | 1:500 |
| Antibody | Anti-mouse- CD38, rat monoclonal | Biolegend | Clone: 90 | 1:200 |
| Antibody | Anti-mouse- CCR7, rat monoclonal | Biolegend | Clone: 4B12 | 1:50 |
| Antibody | GL7, rat monoclonal | Biolegend | Clone: GL7 | 1:500 |
| Antibody | Anti-mouse- CD4, rat monoclonal | Biolegend | Clone: RM4-4 | 1:500 |
| Antibody | Anti-mouse- CD44, rat monoclonal | Biolegend | Clone: IM7 | 1:200 |
| Antibody | Anti-mouse- CXCR5, rat monoclonal | Biolegend | Clone: L138D7 | 1:100 |
| Antibody | Anti-mouse- PD-1, rat monoclonal | Biolegend | Clone: 29 F.1A12 | 1:200 |
| Antibody | Anti-mouse- CXCR3, Armenian hamster monoclonal | Biolegend | Clone: CXCR3-173 | 1:100 |
| Antibody | Anti-mouse- CCR6, Armenian hamster monoclonal | Biolegend | Clone: 29–2 L17 | 1:50 |
| Antibody | Anti-T-bet, mouse monoclonal | Biolegend | Clone: 4B10 | 1:200 |
| Antibody | Anti-GATA3, mouse monoclonal | Biolegend | Clone: 16E10A23 | 1:50 |
| Antibody | Anti-RORγt, mouse monoclonal | Biolegend | Clone: Q31-378 | 1:100 |

*Continued on next page*

*Continued*

| Reagent type (species) or resource | Designation | Source or reference | Identifiers | Additional information |
|---|---|---|---|---|
| Antibody | Anti-mouse- CD45.2, rat monoclonal | Biolegend | Clone: 104 | 1:100 |
| Antibody | Anti-BCL6, mouse monoclonal | Biolegend | Clone: 7D1 | 1:50 |
| Antibody | Anti-mouse-IgG1, rat monoclonal | Biolegend | Clone: RMG1-1 | 1:200 |
| Antibody | Anti-mouse-IgG2a, rat monoclonal | BD | Clone: R19-15 | 1:200 |
| Antibody | Anti-mouse-IgG3, rat monoclonal | BD | Clone: R40-82 | 1:200 |
| Antibody | Anti-mouse-IgE, rat monoclonal | BD | Clone: R35-72 | 1:200 |
| Antibody | Anti-mouse-IgA, rat monoclonal | BD | Clone: C10-1 | 1:200 |

## Study design

This study aims to investigate the memory function of different Tfh subsets in human and mice. cTfh1/2/17 subsets in total and antigen-specific cTfh cells from healthy donors and vaccinees were analysed for phenotypes and kinetics. In vitro generated Tfh1/2/17-like (iTfh1/2/17) cells were analysed for phenotypes and also function after being transferred into recipient mice followed by immunizations. Human cohort samples sizes varied and were guided by previous studies. Mouse sample sizes of three to five per group per time point were used for experiments to detect significant differences between groups while minimizing the use of laboratory animals. Mice were randomly assigned, age and gender matched between groups. The investigators were blinded in collecting raw data from human and mouse samples.

## Human samples

Demographics of human samples were shown in (*Table 2*). Written informed consent was obtained from participants or the parents of children participants according to the ethics approved by human ethics committees of Renji Hospital affiliated to Shanghai Jiao Tong University School of Medicine (KY2019-161), Fourth Military Medical University (KY20163344-1), Tongji Hospital (NCT05009134), Shanghai Children's Medical Centre affiliated to Shanghai Jiao Tong University School of Medicine and Obstetrics and Gynecology Hospital of Fudan University (Kyy2018-6). Whole blood samples from healthy individuals (cTfh phenotyping, N=33; Measles and TT AIM assay, N=20) were collected from Renji Hospital affiliated to Shanghai Jiao Tong University School of Medicine, Shanghai, China. Whole blood samples from healthy volunteers (N=38) who received the standard recombinant HBV vaccine (Shenzhen Kangtai Biological Products Co.) were recruited by Fourth Military Medical University, Xi'an, China. Whole blood samples from healthy volunteers (qPCR and cytokine assay, N=14; Recovered Covid-19 patients, N=13) were collected from Tongji Hospital affiliated to Huazhong University of Science and Technology Tongji Medical College, Wuhan, China. Whole blood samples from children (N=18) were collected from Shanghai Children's Medical Centre affiliated to Shanghai Jiao Tong University School of Medicine, Shanghai, China. Cord blood samples (N=5) were collected from Obstetrics and Gynecology Hospital of Fudan University, Shanghai, China. Buffy coats from healthy donors for bulk RNA-seq were obtained from the blood bank of Changhai Hospital affiliated to Navy Medical University, Shanghai, China.

## Mice

CD45.1 WT, CD45.2 WT, CD28KO, B1-8 and OT-II mice were maintained on a C57BL/6 background and housed in specific pathogen-free conditions in the Australian Phenomics Facility (APF). All animal experiments were carried under protocols (ethics number: A2019/36) approved by ANU's animal ethics committee.

## PBMC and plasma isolation

Blood from human and mouse were collected in BD Vacutainer Blood Collection Tubes. After centrifugation (400 g, 20 °C, 5 min), plasma was collected and stored in –80 °C for further analysis. Blood cells

or buffy coats were diluted in PBS and gently loaded onto the Ficoll-Paque Plus (GE Healthcare) at the volume ratio of 1:1, followed by density gradient centrifugation (450 g, 20 °C, 20 min, no brake). PBMC were then aspirated and resuspended in cold PBS for further experiment.

### Antigen-induced marker assay (AIM assay)

Cryopreserved PBMC were thawed, washed and counted. A total of $5 \times 10^5$ PBMCs were resuspended in 200 μL complete RPMI media (3% FBS for AIM assay) and cultured in a 96-well flat-bottom plate for 18 hr in the presence of 20 μg/mL recombinant HBVSA (Beijing Bioforce), tetanus toxin (Sigma), measles (GenWay) or 1 μg/mL SARS-CoV-2 Prot_S (Miltenyi Biotec), no antigen was added to control wells. At least six wells were seeded (3 antigen treated wells +3 medium only wells) for each PBMC sample. FACS was performed and the replicates for each sample were merged for downstream analysis.

### Quantitative RT-PCR

Total RNA was extracted from sorted T cell subsets using Trizol reagent (Thermo Fisher Scientific) or RNAeasy Micro Kit (Qiagen) and reverse-transcribed to cDNA using PrimeScript RT reagent kit (TaKaRa Biotechnology). RT-PCR was performed with StepOnePlus (Applied Biosystems) using SYBR Green PCR Master Mix (Thermo Fisher Scientific) with specific primers. The reaction of PCR was performed according to the following protocol: 95 °C for 2 min, followed by 40 cycles of 95 °C for 10 s, a specific annealing temperature for 10 s, and 72 °C for 15 s. Relative gene expression was calculated by 2(−Delta Delta CT) method using GAPDH (for human cells) and Ubc (for mouse cells) as an endogenous control.

### ELISA for detecting antibody titer

For detecting anti-HBVSA antibody titer (total binding), a commercialized ELISA kit was used (Shanghai Kehua Bio-engineering Co.). In brief, plasma was diluted 10 times and incubated with HBVSA pre-coated ELISA plate for 30 min under 37 °C. Then HBVSA-HRP was added for another 30 min incubation, followed by five washes and substrate solution was used to determine the OD450 value. The antibody titer was calculated according to the standard curve generated by the standard with a known antibody titer. For detecting anti-HBVSA antibody titer (total IgG), plasma was diluted 10 times and incubated with HBVSA pre-coated ELISA plate for 30 min under 37 °C. Then the plate was washed three times, and added by anti-human IgG-HRP (1:60,000 dilution, Sigma) for 30 min incubation under 37 °C. The plate was then washed five times, and substrate solution was used to determine the OD450 value. IgG specific to SARS-CoV-2 spike (S) and nucleocapsid (N) proteins in plasma were measured using chemiluminescent immunoassay kits (Yhlo Biotech Co) as previously described (*Yao et al., 2022*). For detecting the anti-NP antibody titer, mouse serum was diluted 2000 times and incubated with NP2-BSA or NP23-BSA pre-coated ELISA plate for 1 hr at RT, followed by three washes and incubated with anti-mouse total IgG-HRP antibody for 1 hr at RT. The plate was then washed five times, and TMB chromogen solution was used to determine the OD405 value with 0.1% SDS as the stop solution.

### In vitro survival and proliferation assays

For in vitro apoptosis assay, FACS purified CD4$^+$ T cells were resuspended in complete RPMI media (10% deactivated FBS (v/v), 100 units/mL penicillin, 100 μg/mL streptomycin, 1 mM sodium pyruvate, 1% MEM nonessential amino acids (v/v), and 0.055 mM \beta-Mercaptoethanol in RPMI 1640 with L-glutamine and 25 mM HEPES) and cultured for 3 days, followed by Annexin V and zombie aqua (Biolegend) staining by FACS. For in vitro proliferation assay, FACS purified CD4$^+$ T cells were labelled by 5 μM CFSE (Thermo Fisher Scientific) for 5 min, washed and seeded on 96-well U-bottom plate with T cell activation Dynabeads (Thermo Fisher Scientific) at the ratio of 3:1 (cell number:bead number) to culture for 2.5 days, followed by FACS to determine the fluorescence of CFSE. Division indices were calculated according to the online tutorial by Flowjo.

### T cell stimulation assay

To evaluate the effect of TCR stimulation on CXCR3 and CCR6 expression by Tfh cells, sorted cTfh cell subsets ($2 \times 10^4$/well) were stimulated with plate-bound αCD3 (5 μg/mL) and αCD28 (2 μg/mL)

or rested for 18 hr in complete RPMI media. CXCR3 and CCR6 expression by Tfh cell subsets were analysed by FACS.

## Flow cytometry analysis

Surface staining was conducted by incubating the cells with the antibodies under room temperature for 30 min in FACS buffer (PBS + 2% FBS). For staining of intracellular cytokines, human cells were stimulated with PMA and ionomycin (500 ng/mL, eBioscience) in the presence of GolgiPlus and GolgiStop (BD Biosciences) for 4 hr at 37 °C. After surface staining, cells were permeabilised using Cytofix/Cytoperm (BD Biosciences). Antibodies specific to cytokines were incubated with cells for 30 min at 4 °C. For intranuclear staining, surface staining was performed followed by fix/perm (eBioscience) and stained for nuclear proteins under room temperature for 45 min. Flow cytometry was performed on a FACS analyser (Fortessa X-20, BD) and the data were analyzed by FlowJo (TreeStar).

## RNA-seq data analysis

0.5–1 million naive, $T_{CM}$, $T_{EM}$, $Tfh_{CM}$, and $Tfh_{EM}$ cells were sorted and extracted total RNA was sequenced by the Illumina platform, and the generated pair-end reads were processed online under the Galaxy project according to a standardised pipeline (*Jalili et al., 2020*). The count files were analysed according to a published pipeline (*Law et al., 2016*) for cpm normalization, MDS plot generation and differentially expressed genes calculation (low count genes were removed by filterByExpr). The heatmap was visualized by HemI. Gene set enrichment analysis (GSEA) was performed by fgsea to calculate GSEA *p* value and normalized enrichment score.

## 10x single-cell RNA-seq analysis

R script for this analysis was provided in the supplementary file. In brief, the processed Seurat object was downloaded from GSE152522 and loaded into Seurat package (*Hao et al., 2021*). Unsupervised clustering was then performed to extract CXCR5-expression cTfh clusters. Then Tfh1 and Tfh17 signature scores were calculated for each cell by AddModuleScore function based on the signature gene sets for Tfh1 and Tfh17 derived from GSE123812. For TCR clonality analysis, cells sharing the same TCR alpha and beta chain CDR3 amino acid sequences were assigned to the same clonotype and the clonal abundance was calculated and ranked. Finally, the Tfh1 and Tfh17 signature scores for the abundant TCR clones (abundance ≥10) were extracted for statistical analysis and visualization.

## Cell transfer and immunisation

For adoptive transfer of in vitro differentiated OT-II cells, CD44[+] OT-II cells cultured under iTh0 or iTfh1/2/17 conditions were FACS-purified and 5×10[4] cells were transferred into each CD28KO recipient mice, followed by OVA or NP-OVA in alum immunisations. For immunization, 50 µg ovalbumin (OVA) or NP-OVA was emulsified in alum (volume ratio 1:1) and injected through intraponeal for a dose of 200 µL per mouse. The spleens were collected on day 7 after immunisation or otherwise indicated on the paper.

**Table 3.** Conditions for differentiating iTfh0, iTfh1, iTfh2, and iTfh17 Cells.

| Cell type | Cytokines | Neutralizing antibodies |
|---|---|---|
| iTh0 | No cytokines | No antibodies |
| iTh1 | 20 ng/ml IL-12 | Anti-IL-4, anti-TGF-β |
| iTh2 | 50 ng/ml IL-4 | Anti-IFN-γ, anti-TGF-β |
| iTh17 | 50 ng/ml IL-6, 2 ng/ml TGF-β | Anti-IFN-γ, anti-IL-4 |
| iTfh1 | 100 ng/ml IL-6, 50 ng/ml IL-21, 1 ng/ml IL-12 | Anti-IL-4, anti-TGF-β |
| iTfh2 | 100 ng/ml IL-6, 50 ng/ml IL-21, 20 ng/ml IL-4 | Anti-IFN-γ, anti-TGF-β |
| iTfh17 | 100 ng/ml IL-6, 50 ng/ml IL-21, 0.1 ng/ml TGF-β | Anti-IFN-γ, anti-IL-4 |

## In vitro differentiation for OT-II cells

The method to differentiate iTfh1, iTfh2, and iTfh17-polarized cells in vitro was developed based on our previous paper (*Gao et al., 2020*). In brief, red blood cell lysed splenocytes from WT mice were left untreated (for differentiating iTh0/1/2/17) or pre-treated by 1 µg/mL lipopolysaccharide (LPS) for 24 hr in the complete RPMI media. $5×10^5$ per well LPS pre-treated splenocytes were co-cultured with FACS purified OT-II cells at the ratio of 50:1 in the presence of 1 µg/mL $OVA_{323-339}$ peptide and indicated cytokines (*Table 3*) for 72 hr to differentiate iTfh1/2/17 cells. No cytokines were added for differentiating Th0 cells. Neutralizing antibodies anti-IL-4, anti-IFN-γ, and anti-TGF-β (BioxCell) were used at 10 µg/mL. Cytokines were purchased from PeproTech.

## Statistical analysis

For human result analysis, data were not assumed Gaussian distributed thus comparisons between two groups were performed by two-tailed Wilcoxon matched-pairs signed rank test and multiple comparisons were performed by Friedman test. For mouse result analysis, data were assumed Gaussian distributed thus comparisons between two groups were performed by two-tailed unpaired *t*-test and multiple comparisons were performed by either one-way or two-way ANOVA test as specified in this paper. For all statistics mean ± SD were showed. Corrections were not applied for multiple comparison tests because comparisons in this study were planned with specific hypotheses specified in advance. Statistical analysis was performed by Prism 9.0 software (GraphPad). p-values <0.05 were considered significant.

## Acknowledgements

We thank supports from Harpreet Vohra and Michael Devoy of the Imaging and Cytometry Facility at the Australian National University. The authors thank the funding supports by Australian National Health and Medical Research Council (NHMRC, GNT2009554, GNT2000466), the Bellberry-Viertel Senior Medical Research Fellowship, ANU and UQ Intramural Funding to DY, NHMRC grants (GNT1158404 to IAC, GNT1173871 to KK, and GNT1194036 to THON), the National Natural Science Foundation of China grants (82130030 and 81920108011 to Z.L. and 82101198 to YYao), the National Key Research and Development Program of China (2017YFC0909003) to LLu, and Shandong Provincial Natural Science Foundation (ZR2020ZD41, 2021ZDSYS12) to YW Part of this research was carried out at the Translational Research Institute, Woolloongabba, QLD 4102, Australia. The Translational Research Institute is supported by a grant from the Australian Government. The funders of the study had no involvement in the study design, data collection, data analysis, interpretation, writing of the report, or decision to submit the paper for publication.

## Additional information

### Funding

| Funder | Grant reference number | Author |
|---|---|---|
| National Health and Medical Research Council | GNT2009554 | Di Yu |
| National Natural Science Foundation of China | 82130030 | Yin Yao Zheng Liu |
| National Key Research and Development Program of China | 2017YFC0909003 | Liangjing Lu |
| Natural Science Foundation of Shandong Province | ZR2020ZD41 | Yunbo Wei |
| National Health and Medical Research Council | GNT200046 | Thi HO Nguyen |

| Funder | Grant reference number | Author |
|---|---|---|
| National Health and Medical Research Council | GNT1194036 | Thi HO Nguyen |
| National Health and Medical Research Council | GNT1158404 | Thi HO Nguyen |
| National Natural Science Foundation of China | 81920108011 | Yin Yao Zheng Liu |
| National Natural Science Foundation of China | 82101198 | Yin Yao Zheng Liu |
| Natural Science Foundation of Shandong Province | 2021ZDSYS12 | Yunbo Wei |
| National Natural Science Foundation of China | 82071792 | Yunbo Wei |

The funders had no role in study design, data collection and interpretation, or the decision to submit the work for publication.

## Author contributions

Xin Gao, Conceptualization, Data curation, Software, Formal analysis, Investigation, Writing – original draft, Project administration, Writing – review and editing; Kaiming Luo, Data curation, Formal analysis; Diya Wang, Yunbo Wei, Jun Deng, Data curation, Methodology; Yin Yao, Data curation, Formal analysis, Methodology; Yang Yang, Methodology; Qunxiong Zeng, Xiaoru Dong, Le Xiong, Dongcheng Gong, Kai Pohl, Shaoling Liu, Yu Liu, Lu Liu, Data curation; Lin Lin, Resources; Thi HO Nguyen, Lilith F Allen, Validation; Katherine Kedzierska, Resources, Validation; Yanliang Jin, Mei-Rong Du, Wanping Chen, Resources, Data curation; Liangjing Lu, Resources, Funding acquisition; Nan Shen, Resources, Data curation, Methodology; Zheng Liu, Resources, Supervision, Funding acquisition; Ian A Cockburn, Conceptualization, Resources, Supervision, Funding acquisition; Wenjing Luo, Conceptualization, Resources, Supervision, Funding acquisition, Project administration; Di Yu, Conceptualization, Resources, Supervision, Funding acquisition, Methodology, Writing – original draft, Project administration, Writing – review and editing

## Author ORCIDs

Xin Gao http://orcid.org/0000-0002-5927-2241
Katherine Kedzierska http://orcid.org/0000-0001-6141-335X
Di Yu http://orcid.org/0000-0003-1721-8922

## Ethics

Written informed consent was obtained from participants or the parents of children participants according to the ethics approved by human ethics committees of Renji Hospital affiliated to Shanghai Jiao Tong University School of Medicine (KY2019-161), Fourth Military Medical University (KY20163344-1), Tongji Hospital (NCT05009134), Shanghai Children's Medical Centre affiliated to Shanghai Jiao Tong University School of Medicine and Obstetrics and Gynecology Hospital of Fudan University (Kyy2018-6).

All animal experiments were carried under protocols (ethics number: A2019/36) approved by ANU's animal ethics committee.

## Decision letter and Author response

Decision letter https://doi.org/10.7554/eLife.82217.sa1
Author response https://doi.org/10.7554/eLife.82217.sa2

# Additional files

## Supplementary files

• MDAR checklist

• Source code 1. The code (R) for scRNA-seq. The original R code to generate the signature scores in *Figure 6* and *Figure 6—figure supplement 1*.

## Data availability

Sequencing data have been deposited in GEO under the accession code GSE167309.

The following dataset was generated:

| Author(s) | Year | Dataset title | Dataset URL | Database and Identifier |
|---|---|---|---|---|
| Gao X | 2023 | T Follicular Helper 17 (Tfh17) Cells are Superior for Immunological Memory Maintenance | https://www.ncbi.nlm.nih.gov/geo/query/acc.cgi?acc=GSE167309 | NCBI Gene Expression Omnibus, GSE167309 |

The following previously published datasets were used:

| Author(s) | Year | Dataset title | Dataset URL | Database and Identifier |
|---|---|---|---|---|
| Meckiff B, Suastegui CR, Rosas VF, Chee SJ, Kusnadi A, Simon H, Grifoni A, Pelosi E, Sette A, Ay F, Seumois G, Ottensmeier CH, Vijayanand P | 2021 | Imbalance of regulatory and cytotoxic SARS-CoV-2-reactive CD4+ T cells in COVID-19 | https://www.ncbi.nlm.nih.gov/geo/query/acc.cgi?acc=GSE152522 | NCBI Gene Expression Omnibus, GSE152522 |
| Yost KE, Satpathy AT, Wells DK, Qi Y, Kageyama R, Wang C, Sarin KY, Brown RA, Bucktrout SL, Davis MM, Chang AS, Chang HY | 2019 | Clonal replacement of tumor-specific T cells following PD-1 blockade [bulk RNA] | https://www.ncbi.nlm.nih.gov/geo/query/acc.cgi?acc=GSE123812 | NCBI Gene Expression Omnibus, GSE123812 |

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
