## [Editor Report]

In regard to subpopulations of memory Tfh cells, the authors nicely showed the significance of Tfh17 cells. Although some concerns about their functional advantages relative to other Tfh subpopulations, Tfh1 and Tfh2, were raised by the reviewers, the authors nicely responded to their concerns. This study contributes to understanding the heterogeneity of memory Tfh cells and provides clues for better vaccine designs.

---

## [Decision Letter]

**Decision letter after peer review:**

Thank you for submitting your article "Type 17 Follicular Helper T (Tfh17) Cells are Superior for Memory Maintenance" for consideration by *eLife*. Your article has been reviewed by 3 peer reviewers, and the evaluation has been overseen by a Reviewing Editor and Satyajit Rath as the Senior Editor. The following individuals involved in review of your submission have agreed to reveal their identity: Masato Kubo (Reviewer #1); Joe Craft (Reviewer #2).

Essential revisions:

1) As the reviewer 3 suggested, the Figure 7F should be removed or modified, and the text should be clarified so it is clear that all the human Tfh work should be clearly labelled as cTfh.

2) As all the reviewers request, authors should provide the functional data in regard to clarifying the advantage of Tfh17cm cells in humoral immunity. Do they have an advantage over becoming GC Tfh cells compared to Tfh1 and Tfh2 cm populations, and if so, do they have a different phenotype within the GC, or is part of their effect on pre-GC (CSR)?

*Reviewer #1 (Recommendations for the authors):*

This reviewer personally recommends that the author provide more evidence to indicate the functional relevance of Th17-derived Tfhcm instead of a phenotypical change of chemokine receptor expression.

*Reviewer #3 (Recommendations for the authors):*

The authors need to clearly describe how they define cTfh in the text (p7), as multiple other labs define cTfh in other ways, with the inclusion of markers (ICOS, CD38, PD-1) in addition to CXCR5, as is used here.

Any use of CXCR5+ cells from the blood must be referred to as cTfh, not Tfh, to clearly distinguish these cells from real Tfh cells found in secondary lymphoid organs.

What is the functional relevance of dividing CXCR5+ cells into TfhEM and TfhCM (figure 3)? I'm not sure that more subsets of cTfh cells are required to make the point about Tfh1/2/17 maintenance and recall.

Using CD45- as a surrogate for immunological age is a huge oversimplification and should not be used as it is incorrect. The authors either need to; remove these plots from the manuscript entirely, rename "immunological age" as what is actually is, %CD45RA- of CD4^+^ ensuring that they do not imply that this is a surrogate for immunological age, or use a previously validated method to estimate biological age e.g. an epigenetic clock. This reviewer prefers the first suggestion as it is not clear what this analysis adds, as the %CD45RA- cells does not perform better than age in years in the correlations supplied.

---

## [Author Response]

Essential revisions:1) As the reviewer 3 suggested, the Figure 7F should be removed or modified, and the text should be clarified so it is clear that all the human Tfh work should be clearly labelled as cTfh.

Thank you for your suggestions.

Figure 7F has been removed.The names for all human Tfh populations in blood have been updated as cTfh (e.g., cTfh1/2/17) in the text and in the figures. We also added a table (Table 1) to summarize the definition of all different Tfh populations in our study as reference.

2) As all the reviewers request, authors should provide the functional data in regard to clarifying the advantage of Tfh17cm cells in humoral immunity. Do they have an advantage over becoming GC Tfh cells compared to Tfh1 and Tfh2 cm populations, and if so, do they have a different phenotype within the GC, or is part of their effect on pre-GC (CSR)?

Following the suggestion from the editor and reviewers, we have conducted several new experiments to investigate the advantage of Tfh17 cells as compared to Tfh1 or Tfh2 cells in humoral immunity. Three non-exclusive hypotheses have been tested.

1) To test whether Tfh17 cells can better survive than Tfh1/2 cells, we transferred either OT-II iTfh1, iTfh2 or iTfh17 cells into CD28KO mice and counted the numbers of transferred cells in the spleen after 1 day and 35 days (Figure 8A). While the numbers of transferred iTfh1/2/17 cells were comparable on day1, the numbers of transferred iTfh17 cells were significantly higher than iTfh1 cells on day35 (Figure 8B-C), suggesting that iTfh17 cells had superior survival capacity over iTfh1 but not iTfh2 cells.

2) To test whether Tfh17 cells may maintain better potential to differentiate into GC-Tfh cells after resting, we transferred either OT-II iTfh1, iTfh2 or iTfh17 cells into CD28KO mice together with NP-specific B1-8 cells, followed by an immediate NP-OVA immunization at day 1 or a delayed NP-OVA immunization to examine the formation of GC-Tfh cells (Figure 8D). In the immediate immunization, iTfh1/2/17 cells expanded and differentiated into GC-Tfh in comparable manners after immunization (Figure 8E-G). However, in the delayed immunization (day 35), iTfh17 cells showed higher expansion than iTfh1 but not iTfh2 cells (Figure 8E). Furthermore, iTfh17 cells differentiated into more GC-Tfh cells than both iTfh1 and iTfh2 cells (Figure 8F-G). These results suggest that iTfh17 cells maintained a better potential to generated GC-Tfh cells compared to Tfh1 or Tfh2 cells, in addition to a better survival than iTfh1 cells.

(3) To compare the B cell helper function between iTfh1/2/17-derived GC-Tfh cells, we sorted iTfh1/2/17-derived CXCR5^hi^ PD-1^hi^ GC-Tfh cells in the same experiment as in ‘(2)’ and measured the expressions of key functional genes in GC-Tfh cells including *Pdcd1, Cxcr5, Icos, Cd40lg, Il21* and *Bcl6*. In the delayed immunization. Despite of better B_GC_ and B_ASC_ responses in the iTfh17 group, we found no significant differences in these gene expression among iTfh1/2/17-derived GC-Tfh cells (Figure 8—figure supplement 1A-B). Such results suggest that iTfh1/2/17-derived GC-Tfh cells had comparable B helper function.

In summary, our results suggested that the superior immunological memory maintenance of iTfh17 cells was attributed to their better survival capacity and better maintenance of the potential to differentiate into GC-Tfh cells, rather than better B cell helper function on per cell basis than that of iTfh1 or iTfh2 cells.

Furthermore, we measured the expressions of *IFNγ* and *Il4* in iTfh1/2/17 derived GC-Tfh cells and demonstrated iTfh1 and iTfh2-derived GC-Tfh cells showed featured of increased *IFNγ* and *Il4* respectively, as their counterpart Th1 and Th2 cells (Figure 8—figure supplement 1C). In agreement with polarized cytokine profiles, we detected that iTfh1 cells promoted isotype switching to IgG2a/IgG3 while iTfh2 cells promoted isotype switching to IgG1/IgE (Figure 8—figure supplement 1D). These results suggest iTfh1/2 cells retained polarised cytokine profiles that promote specific class-switch recombination after antigen re-exposure.

We have added the new results and updated the manuscript accordingly (line 348-386).

Reviewer #1 (Recommendations for the authors):This reviewer personally recommends that the author provide more evidence to indicate the functional relevance of Th17-derived Tfhcm instead of a phenotypical change of chemokine receptor expression.

This recommendation is related to this reviewer’s above comment on the relevance of CCR6 expression to Tfh17’s function. It should be noted that the expression of CCR6 was used as a marker to identify Tfh17 subsets and the difference of Tfh17 and Tfh1/2 is broad and beyond CCR6. We have now added more results to investigate the survival and differentiation potential of Tfh17 v.s. Tfh1/2 cells (new Figure 8). Please refer to the response (2) to Essential Revisions for details.

We agree with this reviewer that the function of CCR6 was not examined in our study. In addition to mediate the cell migration into the inflammatory sites, CCR6 also influences the position of lymphocytes in lymphoid organs (*CCR6-dependent positioning of memory B cells is essential for their ability to mount a recall response to antigen,* Journal of Immunology 2015; *Early CCR6 expression on B cells modulates germinal centre kinetics and efficient antibody responses*, Immunology and Cell Biology 2017). The role of CCR6 expression by Tfh17 cells in regulating the supremacy in immunological memory maintenance can be investigated in future study.

Reviewer #3 (Recommendations for the authors):The authors need to clearly describe how they define cTfh in the text (p7), as multiple other labs define cTfh in other ways, with the inclusion of markers (ICOS, CD38, PD-1) in addition to CXCR5, as is used here.

In this study, cTfh cells were defined as CD4^+^ CD45RA^-^ CXCR5^+^ cells (line 70) and are align with the field’s mainstream view. ICOS, CD38, PD-1 are expressed by a subpopulation of cTfh cells (CCR7^lo^PD-1^+^ICOS^+^CD38+ cT_FH_ cells are associated with active T_FH_ differentiation – Yu D et al. *Targeting TFH cells in human diseases and vaccination: rationale and practice*, Nature Immunology 2022).

Any use of CXCR5+ cells from the blood must be referred to as cTfh, not Tfh, to clearly distinguish these cells from real Tfh cells found in secondary lymphoid organs.

Thank you for this suggestion and this has now been corrected throughout the text.

What is the functional relevance of dividing CXCR5+ cells into TfhEM and TfhCM (figure 3)? I'm not sure that more subsets of cTfh cells are required to make the point about Tfh1/2/17 maintenance and recall.

Many studies have reported the existence of CCR7^+^ and CCR7^-^ cTfh cells that assemble conventional T_CM_ and T_EM_ T cell populations respectively. In our initial study, we experimentally demonstrated that early generated cTfh cells were largely all CCR7^-^ cTfh_EM_ cells and the phenotype was converted to CCR7^+^ cTfh_CM_ cells (Figure 1F, He J et al. *Circulating precursor CCR7^lo^PD-1^hi^ CXCR5^+^ CD4^+^ T cells indicate Tfh cell activity and promote antibody responses upon antigen reexposure*, Immunity 2013). Therefore, due to the different kinetics of cTfh_EM_ and cTfh_CM_ generation, the preferential enrichment of cTfh1 in cTfh_EM_ cells and the preferential enrichment of cTfh17 in cTfh_CM_ suggest the difference of cTfh1/2/17 subsets in generation or maintenance. This was an important observation that motivated us to conduct the following investigation.

Using CD45- as a surrogate for immunological age is a huge oversimplification and should not be used as it is incorrect. The authors either need to; remove these plots from the manuscript entirely, rename "immunological age" as what is actually is, %CD45RA- of CD4^+^ ensuring that they do not imply that this is a surrogate for immunological age, or use a previously validated method to estimate biological age e.g. an epigenetic clock. This reviewer prefers the first suggestion as it is not clear what this analysis adds, as the %CD45RA- cells does not perform better than age in years in the correlations supplied.

This comment has been addressed by the response Essential revisions #1.